# De novo synthesized Min proteins drive oscillatory liposome deformation and regulate FtsA-FtsZ cytoskeletal patterns

Elisa Godino[1], Jonás Noguera López[1], David Foschepoth [1], Céline Cleij [1], Anne Doerr[1], Clara Ferrer Castellà[1] & Christophe Danelon[1]*

The Min biochemical network regulates bacterial cell division and is a prototypical example of self-organizing molecular systems. Cell-free assays relying on purified proteins have shown that MinE and MinD self-organize into surface waves and oscillatory patterns. In the context of developing a synthetic cell from elementary biological modules, harnessing Min oscillations might allow us to implement higher-order cellular functions. To convey hereditary information, the Min system must be encoded in a DNA molecule that can be copied, transcribed, and translated. Here, the MinD and MinE proteins are synthesized de novo from their genes inside liposomes. Dynamic protein patterns and accompanying liposome shape deformation are observed. When integrated with the cytoskeletal proteins FtsA and FtsZ, the synthetic Min system is able to dynamically regulate FtsZ patterns. By enabling genetic control over Min protein self-organization and membrane remodeling, our methodology offers unique opportunities towards directed evolution of bacterial division processes in vitro.

[1] Department of Bionanoscience, Kavli Institute of Nanoscience, Delft University of Technology, van der Maasweg 9, 2629HZ Delft, The Netherlands. *email: c.j.a.danelon@tudelft.nl

The construction of a synthetic cell from basic molecular components has become an engaging frontier in bottom-up synthetic biology[1–4]. Creating a synthetic system endowed with life-like features will arguably help us understand the fundamental principles of biological cells, while providing a test bed for the engineering of biotechnologically or medically relevant functionalities. Such a constructive approach towards an elementary cell entails the collaborative operation of essential cellular subsystems based on the extant molecular hardware comprising DNA, RNAs, proteins, and lipids. Similar to its natural counterpart, synthetic cells must be able to self-reproduce, to propagate information and to develop new functionalities, a process that implies replication of the genetic material, biosynthesis of the basal constituents, and division of the compartment.

One possible strategy to divide synthetic cell models is to consider the canonical division mechanism of prokaryotes. In rod-shaped bacteria such as *Escherichia coli*, division is carried out by a multiprotein complex, the divisome, and is tightly regulated in time and space by associated mechanisms[5]. FtsZ, the core component of the divisome, polymerizes and assembles into a contractile ring-like structure called the Z-ring[6]. Symmetrical cell division is ensured by restricting the formation of the Z-ring at mid-cell through different inhibitory mechanisms of FtsZ polymerization. In *E. coli*, this spatial organization is primarily achieved by the Min system, which consists of three proteins encoded by the *minB* operon: MinC, MinD, and MinE[7]. The Min proteins self-organize at the inner surface of the cytoplasmic membrane and dynamically oscillate between the two cell poles[8–11]. These oscillations are driven by a reaction-diffusion mechanism involving MinD and MinE, nucleotide exchange, and transitions between cytoplasmic and membrane-bound states[12]. MinC, an inhibitor of FtsZ polymerization, passively travels along the dynamic MinDE protein pattern. The oscillations impose a time-averaged intracellular gradient of MinC concentration that is minimal in the middle of the cell, where FtsZ polymerizes as an early stage of divisome maturation[13–15].

The in vitro reconstitution of *E. coli* Min protein oscillations has been the focus of many studies since the first report by Loose et al.[16]. Using supported lipid bilayers (SLBs) as a model membrane system, planar waves and rotating spirals of purified Min proteins were observed[16,17]. Key factors influencing the wave parameters were identified, such as the ratio between MinD and MinE levels[16], the flow of the solution[17], the phospholipid composition, and the buffer composition[17]. Going beyond planar membranes, various oscillating patterns have been observed in three-dimensional cell-shaped polydimethylsiloxane containers[18,19], as well as in fully enclosed biological compartments such as microdroplets[20] and giant liposomes[21]. Noteworthily, attempts to encapsulate an active Min system inside liposomes have long faced technical challenges and successful experiments have only been reported recently[21]. A variety of dynamical behaviors has been evidenced, which includes pulsing, circling, and pole-to-pole oscillations[20,21]. Remarkably, in some liposomes and under hypertonic conditions, Min protein oscillations were accompanied by large-scale membrane deformation[21].

Although these experiments unveiled some of the critical parameters governing Min oscillations and, in the latter study, beautifully demonstrated the interplay between membrane-assisted protein self-organization and the mechanical properties of the liposome compartment, it remains unknown whether Min oscillations can functionally be encoded in a DNA-based synthetic cell. Unlike earlier efforts that exclusively relied on purified Min proteins, the present work is motivated by the de novo synthesis and functional self-organization of the Min system inside liposomes. In analogy with information storage and

transfer in biology, we believe that in-liposome protein production from a DNA template is a viable route to build an autonomous, self-replicating synthetic cell.

Herein, we reconstitute oscillatory patterns of cell-free expressed *E. coli* MinD and MinE proteins. The PURE (for Protein synthesis Using Recombinant Elements) system[22] was chosen for its low protease and nuclease activity, and for the well-defined nature of its constituents. Unlike whole cell extracts, the PURE system is devoted to gene expression and is devoid of endogenous proteins (here Min and Min-interacting partners) that may interfere with the cellular function to be reconstituted. After optimizing the conditions for in vitro gene expression with the PURE system[22] and for Min protein activity on SLBs, encapsulation inside cell-sized liposomes is successfully realized. Distinct modes of oscillations are reproduced and extensive membrane deformation coupled to internal Min system dynamics is demonstrated. Cytoskeletal patterns of FtsZ and its natural membrane-anchoring protein FtsA, which is also synthesized in the PURE system, are reconstituted on SLBs and their spatial regulation by cell-free expressed MinDE(C) is investigated.

## Results

**Cell-free expression of Min proteins with the PURE system**. We first verified that cell-free expression of the *E. coli* *minC*, *minD*, and *minE* genes leads to full-length proteins. PURE*frex*2.0 was chosen among the different variants of the commercially available PURE system for its relatively high yield and prolonged expression lifetime in bulk[23] and inside liposomes[24]. The MinC/D/E proteins were synthesized from their respective DNA template and the translation products were first analyzed by SDS-polyacrylamide gel electrophoresis (PAGE). Visualization of synthesized proteins was achieved by co-translational incorporation of fluorescently labeled lysine residues and fluorescence gel imaging (Fig. 1a). Expression of MinC (~25 kDa), MinD (~29 kDa), and MinE (~10 kDa) proteins was confirmed and a main band of expected molecular weight was detected in each case, as well as in MinD/E co-expression reactions (Fig. 1b). To obtain quantitative insights about the amount of synthesized proteins, pre-ran PURE system samples were trypsin-digested (Supplementary Fig. 1) and analyzed by liquid chromatography-coupled mass spectrometry (LC-MS). Specific proteolytic peptides covering different locations from the N- to C-terminal parts were identified in both MinD and MinE proteins, whereas only two MinC-specific peptides were found (Supplementary Table 1 and Supplementary Fig. 2). Their abundance was quantified using purified proteins of known concentrations (Supplementary Table 2 and Supplementary Fig. 3) and was plotted as a function of their position along the protein primary sequence (Fig. 1c and Supplementary Fig. 4). Protein concentration was assessed by considering the most C-terminal peptide. Values of $19 \pm 7\,\mu M$ and $5 \pm 4\,\mu M$ (mean ± SD, five biological replicates) were obtained for MinD and MinE, respectively, when the two genes are co-expressed for 3 h. Such concentrations are sufficient to generate dynamic patterns onto a lipid membrane[16]. An estimation of MinC concentration is ~2.5 μM after 3 h, but no peptide localized close to the C-terminal part could clearly be identified. Kinetics of MinD/E production reveals that most proteins are synthesized within ~4 h co-expression (Fig. 1d). Estimations of the apparent translation rate are $0.09 \pm 0.05\,\mu M$ min$^{-1}$ (MinD) and $0.03 \pm 0.02\,\mu M$ min$^{-1}$ (MinE) (mean ± SD of fitted parameter values, three biological replicates), and the expression lifespans, defined as the time points at which protein production stops, are $307 \pm 63$ min (MinD) and $194 \pm 38$ min (MinE). These values are consistent with previous gene expression kinetics using the PURE system[23]. The concentration ratio of

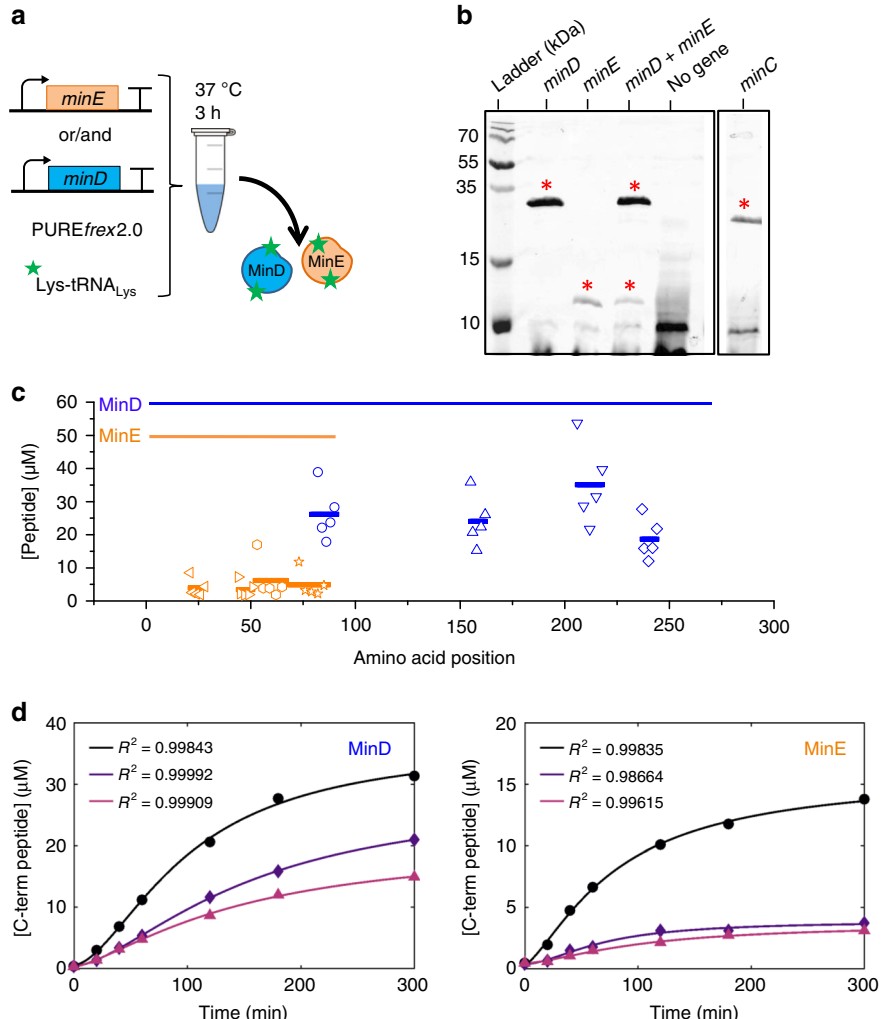

**Fig. 1** Quantification of cell-free gene-expressed MinE and MinD proteins. **a** Schematic of co-translational labeling of MinD and MinE proteins in the PURE system. tRNA_Lys preloaded with a BODIPY-conjugated lysine (GreenLys) is supplemented in a PUREfrex2.0 reaction. Same strategy was applied with MinC. **b** Translation products were analyzed by fluorescence imaging of a 18% polyacrylamide gel. The bands depicted with an upper red star correspond to the full-length protein with expected molecular weight. In a control experiment with no genes added, a smear background of GreenLys is visible, which is distinct from the gene-specific bands. **c** Quantitative LC-MS analysis of MinD and MinE proteolytic peptides and their position along the primary sequence of the proteins. The MinD- and MinE-annotated lines depict the full-length sequences. The amino acid sequence of the different peptides is reported in Supplementary Table 2. Peptide concentrations represent averaged values (displayed as segments whose length matches with the length of the corresponding peptide) over five biological repeats (markers represent individual measurements), of which two included three technical replicates. **d** Concentration of the most C-terminal peptide of MinD (AYADTVER) and of MinE (DGDISILELNVTLPEAEELK) as a function of the expression time course. Symbols represent data from three independent experiments. The solid lines are fits to a mathematical model for gene expression (Equation (1)). Source data are provided as a Source Data file

synthesized MinE and MinD proteins decreases during the first hour of expression and stabilizes to values [MinE]/[MinD] ~ 0.3 between 1 h and 5 h reaction (Supplementary Fig. 5).

**Synthetic MinDE self-organize into dynamic patterns.** In the course of our investigations to optimize activity assays on SLBs with cell-free expressed MinDE proteins, we empirically discovered three critical conditions to reconstitute surface waves. First, PUREfrex2.0 must be supplemented with adenosine triphosphate (ATP, additional 2.5 mM). Presumably, the ATP-dependent MinD dimerization[25] competes with other ATP-consuming processes, e.g., transcription and amino acylation, which reduce the amount of ATP despite the energy recycling enzymatic set present in the PURE system. Second, although the minE gene is directly amplified from the E. coli genome, we found

that sequence optimization for averaged codon usage in E. coli genes, GC content, and avoidance of 5′ RNA secondary structure were necessary. This result may account for the different conditions and limiting reaction steps between cell-free and in vivo gene expression[23]. Third, addition of DnaK mix (a cocktail of highly purified chaperone proteins) during expression enhances protein activity. Under these optimized conditions (Fig. 2a), incubation of co-synthesized MinD and MinE proteins onto an SLB led to the formation of planar waves, rotating spirals, and standing waves (Fig. 2b and Supplementary Movie 1), three typical behaviors of the Min system[16]. Trace amount of purified eGFP-MinD (100 nM) was employed to visualize Min dynamics by spinning disk fluorescence microscopy without directly contributing to wave generation.

Next, we asked whether Min dynamic patterns could be monitored concurrently to protein production directly on top of

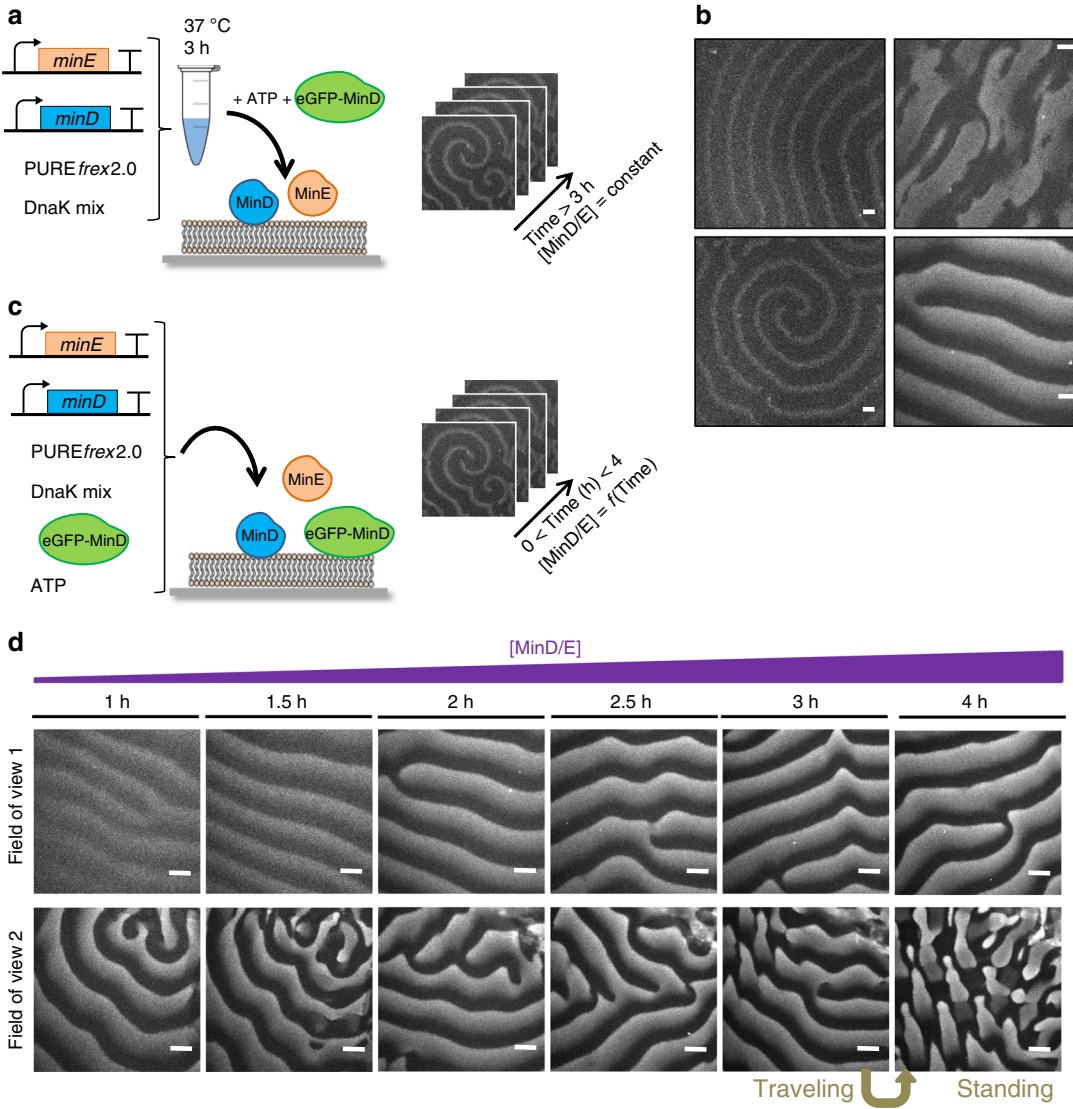

**Fig. 2** Supported membrane assays with de novo synthesized MinD and MinE proteins. **a** Schematic of the experimental workflow for end-point expression assays. Both *minD* and *minE* genes are expressed in a PURE system reaction in the presence of DnaK chaperone mix for 3 h. The solution is then supplemented with 2.5 mM ATP and a trace amount of purified eGFP-MinD (100 nM) before transfer on top of a supported lipid bilayer. **b** Fluorescence microscopy images of representative types of MinDE dynamic patterns. Videos can be found in Supplementary Movie 1. Scale bars are 20 μm. **c** Schematic illustration of the in situ MinDE co-expression and self-organization on an SLB. **d** Several SLB fields of view were imaged at different points during in situ co-expression of MinD and MinE proteins. Corresponding videos are shown in Supplementary Movie 2. Scale bars are 20 μm

the membrane (Fig. 2c). Our gene expression-based approach allowed us to observe in real-time the evolution of the surface waves as a function of MinD/E protein concentrations, providing a decisive advantage over SLB assays conducted exclusively with purified Min proteins. After 1 h expression, the fluorescence intensity was high enough to detect planar waves. As time progresses, the waves become more discernable but their dynamic properties remain constant up to 4 h measurements (Fig. 2d), despite an increase of both MinD and MinE concentrations during this time interval (Fig. 1d). Calculated wavelength is $43 \pm 7$ μm and wave velocity is $0.49 \pm 0.06$ μm s$^{-1}$ (mean ± SD, 26 data points from two biological replicates) (Supplementary Fig. 6). The robustness of Min dynamics over absolute protein concentrations may be explained by the relatively steady ratio of MinE and MinD proteins after 1 h expression (Supplementary Fig. 5). Interestingly, changes between two types of dynamic patterns—from traveling to standing waves—could be tracked (Fig. 2d and Supplementary Movie 2). Standing waves of expressed MinD/E

have a characteristic oscillation time of $47 \pm 4$ s (mean ± SD, 30 data points from two biological replicates). As such transitions were only observed at long time points (>3 h), we hypothesize that a possible cause could be a decrease of ATP concentration. In addition, we performed SLB assays using 1 μM of purified MinD and MinE proteins mixed in a PURE system background. Regular standing and traveling waves with similar properties as synthetic Min patterns were observed (Supplementary Fig. 7 and Supplementary Movie 3). This result indicates that the PURE system itself does not significantly influence Min dynamic properties. Reconstituted surface waves with purified Min proteins in minimal buffers display a wavelength between 25 and 90 μm, and a velocity between 0.1 and 0.4 μm s$^{-1}$[16,17].

**In-liposome-synthesized MinDE proteins oscillate.** After having established a working protocol to produce an active MinDE system in SLB assays, we sought to recapitulate gene expression

and Min oscillations within liposomes. Liposomes were prepared using glass bead-assisted lipid film hydration[24,26]. Compared with microfluidic- and droplet-transfer-based approaches, our method presents the advantage of not containing organic solvent. Moreover, the produced liposomes are smaller (diameter ranging from 1 to 20 µm with an average size around 4 µm), which mimics more closely the size of a bacterial cell than conventional > 20 µm giant vesicles. Two lipid compositions were used (see Methods for full names of the phospholipids): a binary lipid mixture with 1,2-dioleoyl-sn-glycero-3-phosphocholine (DOPC) and 1,2-dioleoyl-sn-glycero-3phosphoglycerol (DOPG) (~3:1 molar ratio) called PC/PG that is commonly used in liposome research and Min reconstitution assays, or a more complex mixture of DOPC, DOPG, 1,2-dioleoyl-sn-glycero-3-phosphoethanolamine (DOPE), and C18:1 cardiolipin (~5:3.6:1.2:0.2 molar ratio) called PC/PG/PE/CL, which emulates more closely the *E. coli* lipid composition. Both types of membranes were doped with biotin–polyethylene glycol- and Texas Red-conjugated lipids. Liposomes enclose PURE*frex*2.0 along with the *minD* and *minE* genes (or one of them in negative controls), DnaK mix, extra ATP and 1 µM purified eGFP-MinC as a fluorescent reporter of the Min oscillations (Fig. 3a). We initially opted for eGFP-MinC in place of eGFP-MinD to avoid interference with lipid film swelling and staining of the outer surface of the liposome membrane. Time-lapse imaging of liposomes at (or close to) their equatorial plane revealed dynamic re-localization of eGFP-MinC between the membrane and the lumen, giving rise to different types of oscillations about 1.5 h after triggering gene expression (earlier time points have not been considered) (Fig. 3b, c). Consistent with previous observations with purified MinDE proteins[21], in situ synthesized Min proteins undergo three prevalent behaviors: pulsing, pole-to-pole, and circling oscillations, with pulsing being the most frequently observed one (Fig. 3 and Supplementary Movies 4 and 5). Liposomes exhibiting Min waving patterns have a diameter spanning the range of 3.5–20 µm, and no clear correlation between liposome size and Min-pulsing frequency was measured (Fig. 3e).

Some liposomes exhibit more complex phenotypes, such as Min waves switching from one oscillation mode to another or to an uncategorized type, halted oscillations, and circling waves that change directionality (Fig. 4a, b and Supplementary Movie 6). The oscillation features can also change over time (Fig. 4c, d) or the membrane signal attenuates while the lumen signal correspondingly increases (Supplementary Fig. 8). All dynamic behaviors described above are unambiguously attributed to successful expression of both functional MinD and MinE, as production of only one of the two proteins fails to reproduce oscillations (Supplementary Movie 7). Sole expression of the *minD* gene leads to stable recruitment of eGFP-MinC to the membrane up to at least 5 h incubation. In contrast, individual expression of MinE results in exclusive localization of eGFP-MinC in the lumen, as expected given that the two proteins do not directly interact.

Moreover, no major differences were observed between the two lipid compositions, both in terms of the oscillation modes and period of Min pulsing (Fig. 3e). It is known that MinD has a higher affinity for anionic lipids[27]. In vitro, it was found that changing the fraction of anionic lipids affects the retention time of MinD and MinE, leading to spatiotemporal modifications in Min patterning[17]. Our results indicate that Min self-organization is robust to the difference in surface charge density between the PC/PG and PC/PG/PE/CL membranes.

In some liposomes, Min oscillations could be monitored for up to 6 h. The time-dependent loss of eGFP-MinC signal at the membrane, or more generally the damping of Min oscillations may indicate the release of MinD as ATP is being consumed,

among other possible scenarios. Indeed, ATP depletion would weaken the ability of MinD to dimerize and bind to the membrane. The fact that not all liposomes exhibit dynamic waves, and potentially also the fact that different types of oscillations are observed, may be explained by different absolute or stoichiometric amounts of synthesized MinD and MinE proteins. Such a functional heterogeneity has been reported for other expressed proteins and is intrinsic to the stochastic partitioning of the many reactants during liposome formation[24,28]. Moreover, we noticed that the success rate of these experiments critically depends on the purity of the DNA template. Although PCR products from the same parental plasmid are prepared with the same reagents according to the same protocol, the number of liposomes displaying Min waves can significantly vary between experiments.

Notably, reconstitution of the complete MinCDE system inside liposomes had not been realized so far. The absolute amount of purified eGFP-MinC (1 µM) is relatively high compared with the concentration used in other studies[29]. Yet, if one assumes similar amounts of synthesized MinD and MinE in liposomes and in bulk reactions, the ratios of [MinC]/[MinE] and [MinC]/[MinD] are ~0.3 and ~0.1, respectively, which is comparable to previous reports[30]. We performed control experiments with a lower concentration of purified eGFP-MinC (0.4 µM), or with 0.2 µM of eGFP-MinD (no MinC). In both conditions, we were able to observe pulsing Min behaviors (Supplementary Fig. 9). However, long time-lapse imaging was challenging due to the low signal intensity.

**Autonomously deforming liposomes by synthetic MinDE proteins**. In a subset of liposomes exhibiting waves, MinDE oscillations are accompanied by membrane remodeling and liposome shape transformation (Fig. 5a and Supplementary Movie 8). When MinD (as deduced from the eGFP-MinC signal) is recruited to the membrane, initially spherical liposomes can elongate and subsequently resume to a sphere as MinD redistributes into the lumen. A similar behavior has recently been reported for deflated vesicles in hypertonic stress conditions[21]. In contrast, no osmotic pressure was externally imposed in the present assay. Although we cannot rule out an osmolarity mismatch between the interior and exterior of some liposomes due to anomalous encapsulation efficiency, it is clear from the apparent sphericity in the non-membrane-bound state of Min proteins that shape deformation is also possible in (near-)isotonic conditions. Of note, when only MinD is expressed, its binding to the membrane (as probed with eGFP-MinC) does not change liposome morphology (Supplementary Movie 7). This observation suggests that transient recruitment of MinE to the membrane actually contributes to liposome deformation.

We further examined membrane deformation in hypertonic conditions by adding 100 mM sucrose to the external medium of the liposomes. Spherical liposomes became deflated exhibiting membrane fluctuations as the Min proteins reorganize (Fig. 5c, Supplementary Fig. 10, and Supplementary Movie 9). When Min oscillations do not take place, liposomes do not periodically change shape (Supplementary Fig. 11). After a few pulsing events, the liposomes relax into a stable sphere even in the presence of a polarized Min protein distribution at the membrane. Remarkably, the force exerted by the Min waves on the inner surface of the liposome membrane, combined with increased membrane flexibility, is sometimes high enough to displace, even to expel, an adjacent vesicle attached to the waving liposome (Fig. 5c and Supplementary Movie 9).

Association of MinD to a lipid bilayer through the formation of an amphipathic helix upon ATP binding is a well-established mechanism[27,31]. The insertion of this helix into the cell

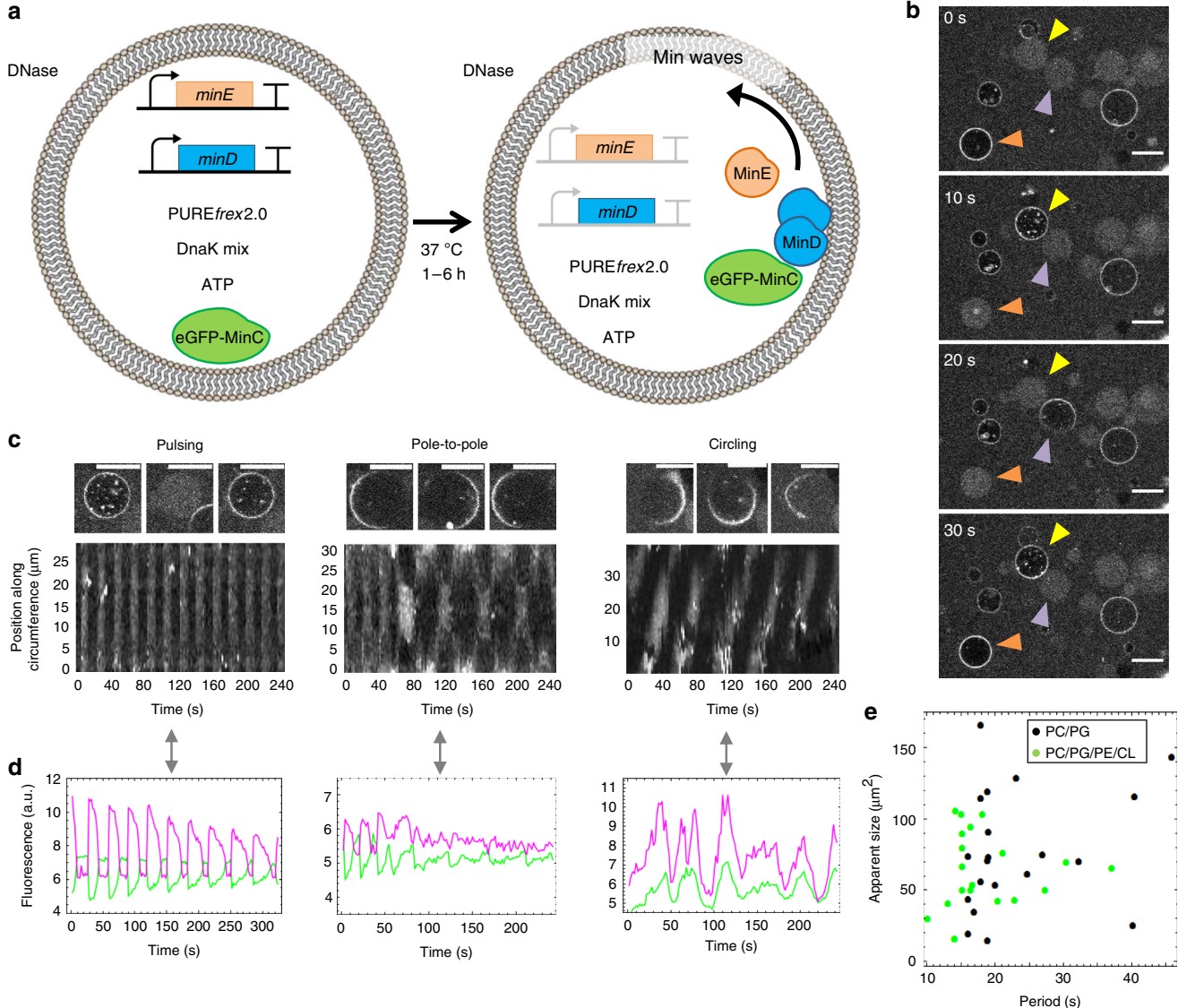

**Fig. 3** In-liposome production and self-organization of MinDE proteins. **a** Schematic illustration of liposome reconstitution assays. The *minD* and *minE* genes are co-expressed within phospholipid vesicles in the presence of DnaK chaperones, extra ATP, and 1 μM of purified eGFP-MinC to report for MinDE dynamics. **b** Time-series images showing liposomes (PC/PG composition) exhibiting periodic redistribution of eGFP-MinC between the membrane and the lumen (Supplementary Movie 4). Colored arrowheads point to Min-active liposomes. Other liposomes show stable fluorescence signal at the membrane or in the lumen throughout the image sequence. Scale bars are 10 μm. **c** Zoomed-in images taken at three different time points (from left to right) of liposomes (PC/PG composition) exhibiting the pulsing, pole-to-pole, or circling modes of oscillations (Supplementary Movie 5). The corresponding kymographs are displayed below. Scale bars are 10 μm. **d** Time traces of the fluorescence intensity of eGFP-MinC at the membrane (magenta) and in the lumen (green) for the three liposomes shown in **c**. a.u., arbitrary unit. **e** The effects of lipid composition and vesicle size have been analyzed for liposomes with a Min-pulsing behavior. Both PC/PG (black dots) and PC/PG/PE/CL (green dots) liposomes show a cross-section area (averaged over the different time point images) that is not obviously correlated with the period of oscillations. Twenty liposomes of each composition have been analyzed

membrane can influence the local curvature and lead to changes in membrane topology. Moreover, it has been reported that MinE can directly interact with the membrane and induce membrane deformation in vitro, in the presence of PG or cardiolipin lipids[31,32]. The interplay between the binding dynamics of MinD and MinE to the membrane and the local variations of membrane curvature remains to be elucidated, in particular the degree to which MinE-induced membrane deformation contributes to the dissociation of MinD from the bilayer. Moreover, the state of membrane permeability to solvent and small solutes, and the effect of an osmolarity differential across the membrane to establish sustained Min oscillations and liposome deformation cycles require further investigation.

**Integration of the synthetic Min and FtsZ systems on SLBs.** Our next goal was to co-reconstitute Min protein patterns and membrane-tethered FtsZ filaments. The third element of the Min system, MinC, interacts with the oscillating MinDE by binding to MinD and it inhibits polymerization of the division protein FtsZ. We first reconstituted membrane-associated FtsZ filaments by anchoring purified FtsZ-Alexa647 to an SLB through the membrane-binding protein FtsA synthesized in the PURE system. Large areas of the SLB were covered with ring-like structures of FtsA-FtsZ copolymers, consistent with previous observations using purified FtsA[33]. MinC and MinD were separately expressed in a test tube and the pre-ran PURE*frex*2.0 solutions were either mixed together or individually mixed with a PURE*frex*2.0 sample

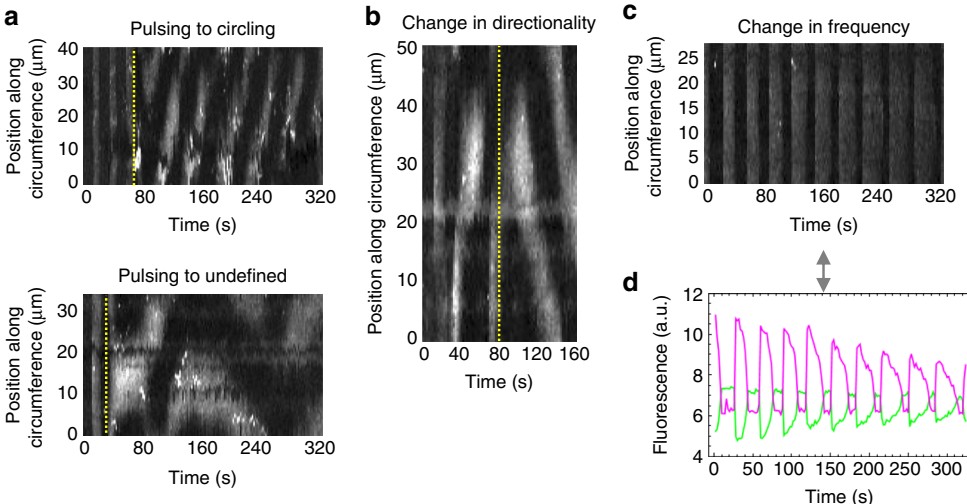

**Fig. 4** Transitions of Min oscillating patterns in liposomes. **a** Kymographs of liposomes exhibiting transitions from the pulsing oscillation mode to a different behavior. **b** Kymograph of a liposome showing circling Min dynamics that changes directionality. In **a** and **b**, the vertical dotted lines depict the transition between two types of oscillations. **c** Kymograph of a liposome with pulsing Min waves whose peak sharpness, frequency, and amplitude decrease over time. **d** Fluorescence time traces of the eGFP-MinC signal localized at the membrane (magenta) or in the lumen (green). The same liposome as in **c** was analyzed. All corresponding videos are shown in Supplementary Movie 6

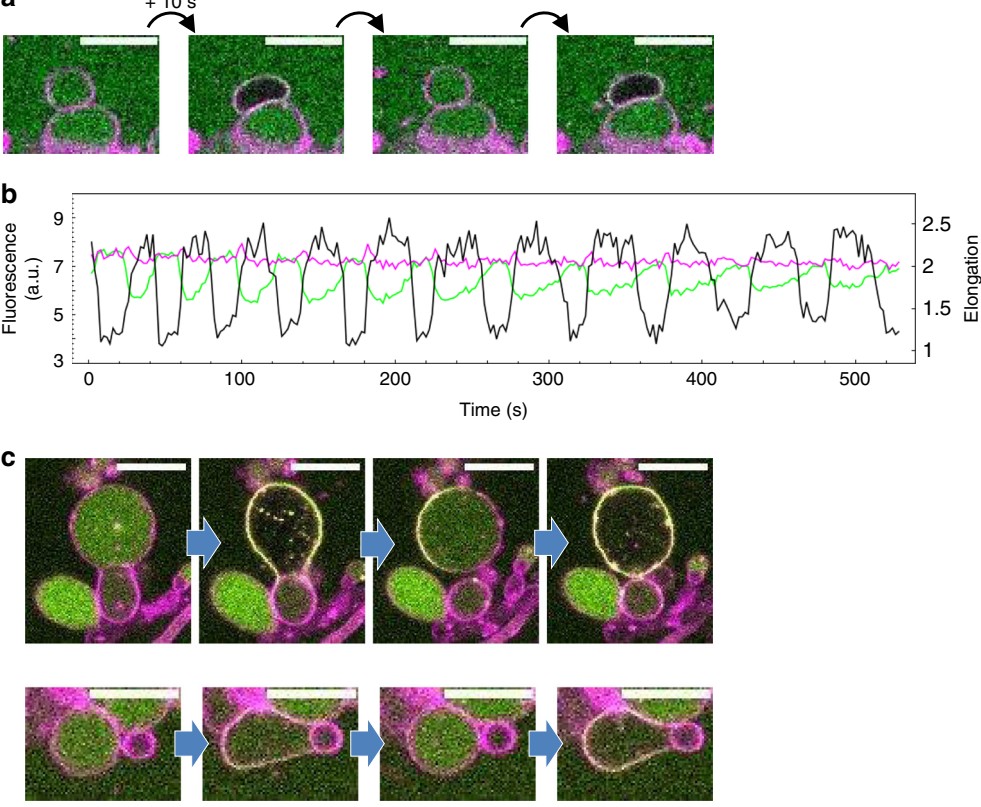

**Fig. 5** Autonomous liposome deformation induced by synthetic Min oscillations. **a** Time-series images of a liposome that changes morphology in response to Min oscillations. The membrane dye signal is colored in magenta and the eGFP-MinC reporting the waves is in green. The corresponding movie is shown in Supplementary Movie 8. Scale bars are 10 µm. **b** Fluorescence time traces of the eGFP-MinC intensity localized at the membrane (magenta) and in the lumen (green) of the liposome displayed in **a**. The time profile of the liposome elongation is overlaid in black and is phase shifted with respect to the amount of eGFP-MinC in the lumen. **c** Time-lapse fluorescence images of two liposomes exposed to a hypertonic external medium (100 mM sucrose). Liposomes undergo dramatic shape deformation upon recruitment of Min proteins to the inner surface of the lipid bilayer. The force generated by the redistribution of the Min proteins is sufficient to push a membrane septum and remodel adjacent vesicles. The corresponding videos are shown in Supplementary Movie 9. Scale bars are 10 µm

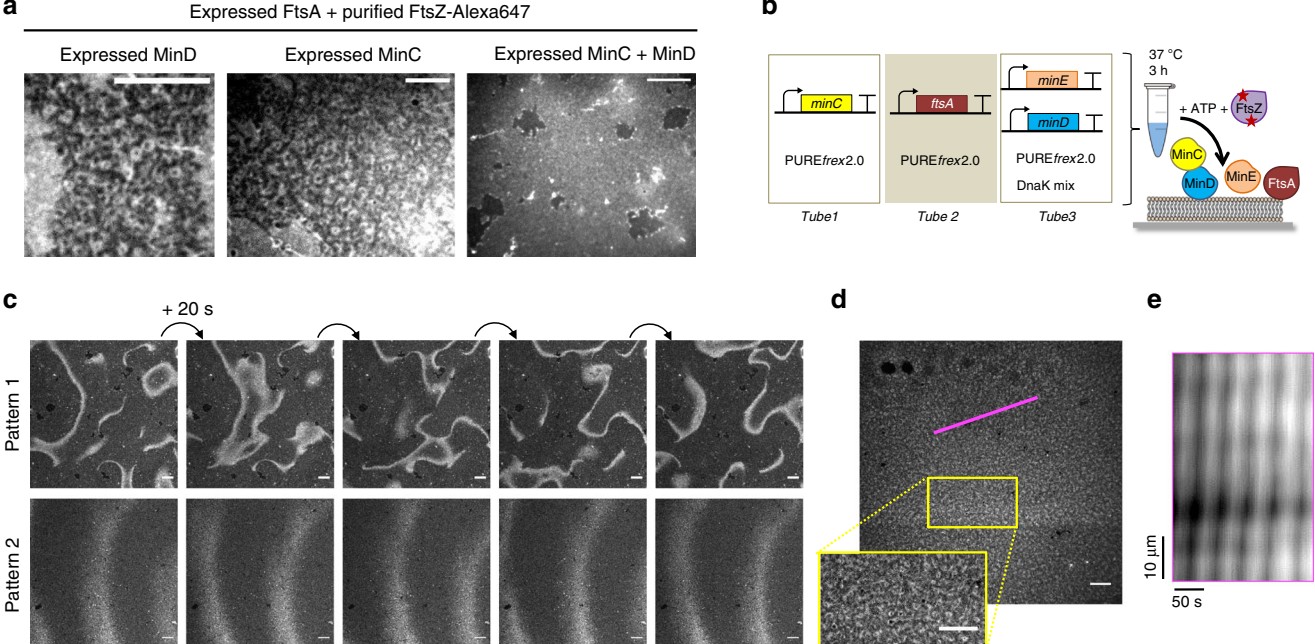

**Fig. 6** Cell-free expressed MinDE(C) regulates FtsZ spatial organization. **a** Ring-like structures composed of purified FtsZ-Alexa647 and expressed FtsA can assemble on an SLB. The ability of separately expressed MinC and MinD, or co-expressed MinC/D to inhibit formation of FtsA-FtsZ rings was investigated. A representative fluorescence image is shown for each condition. An 8 × 8 fields of view montages are shown in Supplementary Figs. 12–14. Scale bars are 10 μm. **b** Schematic of the MinCDE and FtsA-FtsZ-coupled SLB assay. **c** Two time-series images of FtsZ-Alexa647 dynamic patterns. Scale bars are 10 μm. The corresponding videos are shown in Supplementary Movie 10. An 8 × 8 fields of view montage of pattern 1 is shown in Supplementary Fig. 15. **d** Cytoskeletal pattern of expressed FtsA and purified FtsZ-Alexa647 combined with MinDE. In the absence of MinC, the FtsZ bundle network is maintained but traveling waves of FtsZ-Alexa647 are also visible (see Supplementary Movie 11). Scale bars are 10 μm. **e** Kymograph of the FtsZ-Alexa647 signal along the magenta line shown in **d**. The images were preprocessed to filter out small-scale features (Methods)

containing a mock expressed protein (the phi29 terminal protein, TP) to account for volume dilution. With both samples containing solely MinC or MinD, FtsA-FtsZ bundles and rings were clearly visible, although the SLB coverage was much less with MinC (Fig. 6a and Supplementary Fig. 12 and 13). In contrast, addition of MinC plus MinD impaired formation of rings and only unresolved FtsZ structures, presumably monomers or short oligomers, remained recruited to the membrane (Fig. 6a and Supplementary Fig. 14). These results confirm that MinD, by increasing the local concentration of MinC at the membrane, reduces the effective MinC concentration to inhibit FtsZ polymerization.

The complete synthetic MinCDE system was co-reconstituted with the expressed FtsA and purified FtsZ-Alexa647, the latter acting as the fluorescence readout in this assay (Fig. 6b). Large-scale dynamic FtsZ patterns with calculated wavelength and velocity values of $34 \pm 8$ μm and $1.1 \pm 0.3$ μm s$^{-1}$ (mean ± SD, three different fields of view from one sample) (Fig. 6c, pattern 1), or $115 \pm 42$ μm and $2.9 \pm 0.8$ μm s$^{-1}$ (mean ± SD, 12 kymographs from 2 biological replicates) (Fig. 6c, pattern 2), were observed, confirming the full activity spectrum of the in vitro-synthesized MinC. In the absence of MinC, two types of FtsZ-Alexa647 patterns coexist: dim propagating waves with wavelength $152 \pm 53$ μm and velocity $2.8 \pm 0.9$ μm s$^{-1}$ (mean ± SD, 12 kymographs from one sample) that presumably anticorrelate with MinDE waves[34], and a globally stable cytoskeletal network of FtsA-FtsZ ring-like structures (Fig. 6d, e and Supplementary Movie 11). This observation suggests that bundles of FtsA-FtsZ are less affected by MinDE waves than short filaments that are more transiently anchored to the bilayer, hence more susceptible to be outcompeted by the membrane-binding MinD. Moreover, our data indicate that FtsA-FtsZ cytoskeletal patterns might influence

MinDE wave properties by increasing both the wavelength and velocity. Interestingly, FtsZ bundles comprising the FtsZ-YFP-membrane-targeting sequence protein for membrane binding were unaffected by purified MinDE[30], suggesting that FtsA-FtsZ co-filaments are more responsive to the Min system. The role of FtsA as an FtsZ-depolymerizing factor in regulating the composite behavior observed here (Fig. 6d and Supplementary Movie 11) remains to be elucidated.

These results demonstrate for the first time that coupling Min system dynamics with FtsZ and the membrane-associated FtsA is possible. A future challenge will consist in co-expressing all five proteins inside liposomes. This will require the concatenation of multiple genes on a single DNA template to avoid heterogeneous encapsulation of the different DNA molecules. Moreover, the productivity of the PURE system and ribosome processivity will have to be improved to synthesize sufficient amounts of all full-length proteins.

## Discussion

Assembly of a synthetic biological cell from elementary building blocks is arguably one of the biggest and most exciting challenges facing science today. Recent methodological advances in liposome preparation and high-throughput analysis[24], and in reconstitution of functional modules, such as transcription–translation[22], DNA replication[28], phospholipid synthesis[35,36], and light-activated energy production[37], represent encouraging steps towards the construction of a fully autonomous artificial cell. Yet, the implementation of a DNA-encoded liposome division machinery remains a major milestone.

In summary, we showed in the present study that MinD and MinE proteins expressed from their genes can self-organize into

planar waves on SLBs or oscillate between the membrane and lumen when enclosed inside the liposomes. Global shape transformations and local membrane undulations can result from dynamic redistribution of the in vesiculo-synthesized Min proteins. This reversible and autonomous Min-assisted membrane deformability represents one of the few examples of active vesicles[38]. Conceptually, a DNA-centered approach to build cellular functions is less pragmatic than merely relying on purified proteins. Yet, it adheres with our framework to create a synthetic cell from a minimal genome[24,28,39]. Three pillars of cellular life were reconstituted in the present study: (i) a lipid compartment as the evolutionary unit, (ii) information storage in the form of DNA and its decoding with the PURE system to produce proteins, and (iii) protein self-organization mediated by nucleotide exchange, ATP hydrolysis, and transient membrane binding, a hallmark in biological systems. Moreover, the observed membrane-remodeling ability of the synthesized Min proteins imposes less constrains in terms of lipid composition and temperature than alternative mechanisms reported earlier[40–44]. Therefore, one could envision to exploit Min waves to stimulate division by budding in a liposome-based synthetic cell.

Endowing the Min biochemical network with genetic control is uniquely suited to investigate the evolution of pattern dynamics and membrane deformation as a function of protein concentration and synthesis rate, in real-time. In general, our methodology is instrumental to further apprehend the interplay between genetic and nongenetic factors involved in the bacterial division processes. Furthermore, the PURE system should be harnessed to co-express MinDEC with FtsZ and its membrane-anchoring proteins. Min oscillations would then drive both liposome elongation and FtsZ ring positioning. Tuning the absolute and relative protein abundance to ensure proper functioning of the two subsystems could be achieved by modulating the strength of transcription and translation of the individual genes. Additional temporal control over protein production could be realized by implementing transcriptional delay or feedback loops[45,46]. Such a rational engineering approach to endow liposomes with a division mechanism might be complemented with an evolutionary route, whereby a liposome population randomly encloses mutagenized min genes from a library. Those liposomes exhibiting a desirable phenotype, for instance pole-to-pole oscillations or membrane constriction, would be selected and their DNA variants amplified for further rounds of directed evolution.

## Methods

**Purified proteins.** Proteins eGFP-MinD and eGFP-MinC were purified according to published protocols[29]. Protein concentrations were determined in a Bradford assay and by measuring eGFP absorbance. Purified MinE used for MS was a kind gift from the Cees Dekker lab[19]. Purified FtsZ-Alexa647 (kindly provided by Germán Rivas lab) was stored in a buffer containing 50 mM Tris, 500 mM KCl, 5 mM MgCl₂, and 5% glycerol at pH 7 as previously described[47].

**Preparation of DNA constructs.** minD and minE gene fragments were amplified by standard PCR with Phusion High Fidelity DNA polymerase (Thermo Fisher Scientific) from chromosomal E. coli MG1655 (K12) DNA with primers ChD511 and ChD382 (minD), and ChD512 and ChD384 (minE) (Supplementary Table 3). These primers contain overhangs for the pET11-a backbone plasmid. The PCR products and linearized backbone pET11-a DNA were checked on a 1% agarose gel stained with SYBR safe, imaged with a ChemiDocTM Imaging System (BioRad Laboratories), and purified with Wizard SV Gel kit (Promega). The purified DNA was incubated with DpnI (New England BioLabs®, Inc.) to remove residual plasmid and the linear DNA was purified again with Wizard SV Gel kit. DNA concentration and purity were measured using a ND-1000 UV-Vis Spectrophotometer (Nanodrop Technologies). Gibson assembly (Gibson Assembly® Master Mix of New England BioLabs®, Inc.) was performed at equimolar concentrations of linearized plasmid (pET11-a) and DNA fragments for 1 h at 50 °C. Transformation of the Gibson assembly products into E. coli TOP10 competent cells was done by heat shock, after which cells were resuspended in 200 μL of fresh prechilled liquid lysogeny broth (LB) medium and incubated for 1 h at 37 °C and 250 r.p.m. Then, the cultures were plated in solid LB medium with ampicillin and grew overnight at

37 °C. Colonies were picked up and cultured in 1 mL of liquid LB medium with 100 μg mL⁻¹ of ampicillin for 16 h at 37 °C and 250 r.p.m. Plasmid purification was performed using the PureYield™ Plasmid Miniprep System (Promega). The phi29 TP-coding DNA construct was prepared by PCR as previously described[28]. Concentration and purity of DNA were checked on NanoDrop. All sequences were confirmed by sequencing. MinE-, MinC-, and FtsA-coding DNA fragments (starting with a T7 promoter and ending with a T7 terminator) with optimized sequence for codon usage, CG content, and 5′ mRNA secondary structures were inserted in a pUC57, pCC1, and pJET1 plasmid (GeneScript), respectively. Plasmids were amplified and purified as described above.

Linear DNA constructs were prepared by PCR from a parental plasmid using the forward and reverse primers ChD709 and ChD757, respectively, annealing to the T7 promoter and T7 terminator sequences (Supplementary Table 3). Amplification products were checked on a 1% agarose gel and were further purified using Wizard SV gel (standard column protocol). Concentration and purity were measured by NanoDrop. Sequences of the linearized constructs can be found in the Supplementary Methods.

**Cell-free gene expression in bulk.** PUREfrex2.0 (GeneFrontier Corporation, Chiba) was utilized following storage and handling instructions provided by the supplier. Linear DNA constructs were added at a concentration of 5 nM for gel analysis. In co-expression reactions, minD and minE DNA templates were included at 4 nM and 8 nM, respectively, and the solution was supplemented with 1 μL of DnaK Mix (GeneFrontier Corporation). DnaK Mix consists of highly purified E. coli DnaK, DnaJ, and GrpE chaperone proteins. In single-gene expression reactions of minC, ftsA, and TP, the DNA template was included at 5 nM. Gene expression reactions were carried out in 20 μL volume in PCR tubes for 3 h at 37 °C.

**Labeling of in vitro synthesized proteins and gel analysis.** PUREfrex2.0 reaction mixture was supplemented with 0.5 μL of GreenLys reagent (FluoroTect™ Green-Lys, Promega) and gene expression was performed in a test tube as described above. Sample was treated with RNase (RNaseA Solution, Promega) for 30 min and proteins denatured for 10 min at 90 °C in 2× SDS loading buffer with 10 mM dithiotreitol (DTT). Samples were loaded on a 18% SDS-PAGE gel. Visualization of the fluorescently labeled protein was performed on a fluorescence gel imager (Typhoon, Amersham Biosciences) using a 488 nm laser and a band-pass emission filter of 520 nm.

**Quantitative proteomics.** A targeted proteomics approach was used to determine the concentration of cell-free synthesized MinC and co-synthesized MinD and MinE proteins. For the MinD/E co-expression, a kinetic experiment (three independent repeats) was performed to determine protein concentrations at 0, 20, 40, 60, 120, 180, and 300 min. In addition, end-point measurements (two independent repeats) were performed by taking the samples after 180 min. Samples of 5 μL of pre-ran PUREfrex2.0 were incubated at 90 °C for 10 min in 22.6 μL of 27.56 mM Tris-HCl pH 7.6, 4.5 mM DTT, and 1.1 mM CaCl₂. Another 2.5 μL and 1.25 μL of the pre-incubated PUREfrex2.0 solution were diluted to a final volume of 5 μL giving a 2× and 4× dilution of the expressed proteins, respectively. Then, 15.52 mM final concentration of iodoacetamide was added and the solution was incubated for 30 min in the dark. The iodoacetamide reaction was quenched by addition of 4.2 mM final concentration of DTT. Finally, 0.625 μg of Trypsin was added and the solution was incubated overnight at 37 °C. The following day, 2.52 μL of 10% trifluoroacetic acid was added, the sample was incubated at room temperature for 5 min, the solution was centrifuged at 16,200 r.c.f. for 30 min, and the supernatant was transferred to an high-performance liquid chromatography vial for MS analysis.

MS analysis of tryptic peptides was conducted on a 6460 Triple Quad LC-MS system (Agilent Technologies, USA). From the samples prepared as described above, 10 μl were injected to an ACQUITY UPLC® Peptide CSH™ C18 Column (Waters Corporation, USA). The peptides were separated in a gradient of buffer A (25 mM formic acid in MilliQ water) and buffer B (50 mM formic acid in acetonitrile) at a flow rate of 500 μL min⁻¹ and at a column temperature of 40 °C. The column was equilibrated with 98:2 ratio of buffer A to B. After injection over 20 min, the ratio was changed to 75:25 of buffer A to B after which, within 30 s, the ratio went to 20:80 of buffer A to B and was held for another 30 s. Finally, the column was flushed for 5 min with 98:2 of buffer A to B.

Supplementary Table 1 shows the transitions of the MS/MS measurements that were observed in every experiment. EF-Tu is a constant component of the PURE system and we used its proteolytic peptide TTLTAAITTVLAK as an internal standard for variations during sample handling. All proteomics data were analysed in Skyline-daily 4.1.1.18179 (MacCoss Lab, University of Washington, USA). Retention time was predicted after standard runs with the method described above using the Pierce™ Peptide Retention Time Calibration Mixture (Catalog number 88320, Thermo Scientific, USA).

Purified proteins eGFP-MinD and MinE with stock concentrations of 64 μM and 89 μM, respectively, and purified eGFP-MinC with a stock concentration of 127 μM were used as standards for quantitative LC-MS. The MinD and MinE proteins were mixed and a serial dilution was prepared in a buffer containing 20 mM HEPES pH 7.6, 180 mM potassium glutamate and 14 mM magnesium

glutamate. Three dilution series of 10, 5, 2.5, 1.25, and 0.625 µM were prepared independently of each other and were treated according to the same digestion protocol as described above for the PURE*frex*2.0 samples. Samples of 10 µL from two dilution rows were injected from the lowest concentration to the highest with blank measurements between every standard row. Then, the three dilutions of two biological replicates of PURE*frex*2.0 reactions were injected with the lowest concentration first. The third standard row was injected last. Three MinC standard rows were prepared in a similar manner but with the highest concentration being 30 µM of purified protein, which was pipetted individually. A 20 µM standard was prepared independently and was used for the serial dilution giving concentrations of 20, 10, 5, 2.5, 1.25, and 0.625 µM. MinC samples in PURE system were prepared and were treated the same way as the MinD and MinE samples. Expression reactions with MinC (four independent experiments) were run for 3 h at 37 °C. MS data were analyzed in Skyline-daily as mentioned above and integrated peak intensities were plotted in OriginPro 2015 (b9.2.257, OriginLab Corporation, USA). The plotted concentrations of each peptide were fitted with Instrumental variance weighting using the formula $w_i = 1/\sigma_i^2$, where $\sigma_i$ is the error bar size (SD from the average) of each value. The scatter plots with linear regression are shown in Supplementary Fig. 3 and the extracted parameters are listed in Supplementary Table 2.

**Phenomenological fitting of MinD/E production kinetics.** Measured concentrations of MinD/E most C-terminal peptides were fit with a phenomenological model to estimate the main kinetic parameters, such as the final yield, production rate, and translation lifespan (or time to plateau). The following sigmoid equation was used[24]:

$$y = k' + k \frac{t^n}{t^n + K^n} \qquad (1)$$

where $t$ is the time in minutes, $y$ the MinD/E peptide concentration at a given time point, and $k'$, $k$, $K$, and $n$ are fit parameters. Using this expression, the final yield corresponds to $k$ and the plateau time, or expression lifespan, is expressed as:

$$T_{\mathrm{plateau}} = \frac{2K}{n} + K \qquad (2)$$

The apparent translation rate, which is defined as the steepness at time $t = K$, is:

$$v_{\mathrm{translation}} = \frac{kn}{4K} \qquad (3)$$

Kinetics obtained from three independent experiments were fit (Fig. 1d) and the parameter values are reported as the average and SD.

**Fabrication and cleaning of the imaging chamber.** Home-made glass chambers were used in both SLB and liposome experiments. Three microscopy glass slides were glued on top of each other with NOA 61 glue (Norland Products). Holes with a diameter of 2.5 mm were drilled across the slides and then covered on one side by gluing a 150 µm-thick coverslip (Menzel-Gläser) with NOA 61, creating the bottom of the glass chamber. Cleaning was performed by successive washing steps of 10 min each in a bath sonicator (Sonorex Digitec, Bandelin) as follows: chloroform and methanol (volume 1:1), 2% Hellmanex, 1 M KOH, 100% ethanol, and MilliQ water. For SLB experiments, the glass chambers were further treated every one to two experiments with Acid Piranha cleaning.

**Lipids.** DOPC, DOPE, DOPG, 1′,3′-bis[1,2-dioleoyl-sn-glycero-3-phospho]-glycerol (18:1 CL), and 1,2-distearoyl-sn-glycero-3-phosphoethanolamine-N-[biotinyl(polyethylene glycol)-2000 (DSPE-PEG-biotin) were from Avanti Polar Lipids. Texas Red 1,2-dihexadecanoyl-sn-glycero-3-phosphoethanolamine (DHPE-Texas Red) was from Invitrogen.

**Preparation of small unilamellar vesicles.** Small unilamellar vesicles (SUVs) were used as precursors for SLB formation. Lipids DOPC (4 µmol) and DOPG (1 µmol) dissolved in chloroform were mixed in a small glass vial. A lipid film was deposited on the wall of the vial upon solvent evaporation through a gentle flow of argon and was further desiccated for 30 min at room temperature. The lipid film was resuspended with 400 µL of SLB buffer (50 mM Tris, 300 mM KCl, 5 mM MgCl$_2$, pH 7.5) and the solution was vortexed for a few minutes. The final lipid concentration was 1.25 mg mL$^{-1}$. A two-step extrusion (each of 11 passages) was carried out using the Avanti mini extruder (Avanti Polar Lipids) equipped with 250 µL Hamilton syringes (Avant Polar Lipids), filters (drain disc 10 mm diameter, Whatman), and a polycarbonate membrane with pore size 0.2 µm (step 1) or 0.03 µm (step 2) (Nuclepore track-etched membrane, Whatman).

**Formation of supported lipid bilayers.** The imaging chamber was treated with oxygen plasma (Harrick Plasma basic plasma cleaner) for 30 min to activate the glass surface. Immediately after plasma cleaning, the SUV solution was added to the sample reservoir at a final lipid concentration of 0.94 mg mL$^{-1}$ and CaCl$_2$ was added at a final concentration of 3 mM. The chamber was closed by sticking a coverslip using a double-sided adhesive silicone sheet (Life Technologies) and the

SLB was formed during 30 min incubation at 37 °C. Next, the chamber was opened and the SLB was carefully washed six times with SLB buffer.

**Activity assays on supported membranes.** A PURE*frex*2.0 reaction mixture was assembled as described above, to express the *minD* and *minE* (optimized or non-optimized sequences) genes. Unless specified otherwise, the sequence-optimized *minE* construct was used in SLB and liposome experiments. The *minE* and *minD* DNA constructs were added at a final concentration of 4 nM and 8 nM, respectively, in both single-gene and double-gene expression reactions. In single-gene expression reactions with the sequence-optimized *minC* and sequence-optimized *ftsA* genes, the constructs were added at a final concentration of 5 nM. The solution was supplemented with (final concentrations indicated): 2.5 mM ATP, 100 nM purified eGFP-MinD, DnaK (GeneFrontier Corporation), and MilliQ water to adjust the final volume to 20 µL. Following 3 h incubation at 37 °C, the solution was added on top of an SLB and the chamber was sealed by sticking a 20 × 20 mm coverslip with a double-sided adhesive silicone sheet.

For in situ expression of MinD/E proteins, the *minE* and *minD* DNA templates were added at a final concentration of 4 nM and 8 nM, respectively. A 20 µL PURE*frex*2.0 mixture was supplemented with 2.5 mM ATP, 100 nM purified eGFP-MinD, and DnaK mix, and the solution was incubated on top of an SLB. The chamber was sealed as described above and the sample was immediately visualized on a laser scanning confocal microscope up to 4 h of expression. Several fields of view were imaged at different time points throughout the incubation period.

For assays involving FtsZ, the *ftsA*, *minC*, and *TP* genes were individually expressed in a test tube, whereas *minE* and *minD* were co-expressed. After 3 h incubation at 37 °C, 5.5 µL aliquots of the appropriate reaction samples were mixed together with 2 mM GTP, 2.5 mM ATP, and 3 µM-purified FtsZ-A647 (all final concentrations) in a total volume of 20 µL. The mixture was added on top of an SLB and the chamber was sealed before imaging.

Planar membrane assays with purified proteins were performed by adding on an SLB a reaction mixture containing 1 µM eGFP-MinD, 1 µM MinE, 2.5 mM ATP, and, as a background medium, a pre-ran PURE*frex*2.0 solution with the expressed TP protein.

**Spinning disk microscopy.** Min activity assays on SLB were performed using spinning disk imaging with an Olympus iX81 inverted fluorescence microscope equipped with a ×100 oil-immersion objective (Olympus), an iXon3 EMCCD camera (Andor Technology), and a Nipkow spinning disk (CSU-XI, Yokogawa). Fluorescence of eGFP-MinD was measured using a 491 nm laser line and appropriate emission filters (525/50 nm). The software Andor IQ3 (Andor Technology, Ltd) was used for image acquisition and identical settings were used for all experiments. Experiments were conducted at 37 °C using an OKO lab incubator box.

**Preparation of lipid-coated beads.** Two different lipid mixtures were utilized. Composition 1 contains DOPC (50 mol%), DOPE (36 mol%), DOPG (12 mol%), 18:1 CL (2 mol%), DSPE-PEG-biotin (1 mass%), and DHPE-Texas Red (0.5 mass%) for a total mass of 2 mg. Composition 2 contains DOPC (75 mol%), DOPG (25 mol%), DSPE-PEG-biotin (1 mass%), and DHPE-Texas Red (0.5 mass%) for a total mass of 5 mg. Lipids dissolved in chloroform were mixed in a round-bottom glass flask. Methanol containing 100 mM rhamnose was added to the lipid solution in a chloroform-to-methanol volume ratio of 2.5:1. Then, 1.5 g of 212–300 µm glass beads (acid washed, Sigma Aldrich) was poured to the lipid-rhamnose solution and the organic solvent was removed by rotary evaporation at 200 mbar for 2 h at room temperature, followed by overnight desiccation. Lipid-coated beads were stored under argon at –20 °C until use.

**Production and immobilization of gene-expressing liposomes.** A PURE*frex*2.0 reaction mixture was assembled in a test tube as described above. Either or both *minD* and *minE* DNA constructs were added at a final concentration of 4 nM and 8 nM, respectively. The solution was supplemented with (final concentrations indicated) the following: 2.5 mM ATP, 1 µM purified eGFP-MinC, DnaK mix, and MilliQ water to reach a final volume of 30 µL. Whenever specified, 1 µM purified eGFP-MinC was substituted with 0.4 µM of eGFP-MinC or 0.2 µM eGFP-MinD. About 25 mg of lipid-coated beads was transferred to the PURE*frex*2.0 solution and liposomes were formed by natural swelling of the lipid film for 2 h on ice, protected from light. During incubation, the tube was gently rotated manually a few times. Four freeze–thaw cycles were then applied by dipping the sample in liquid nitrogen and thawing on ice. The sample reservoir of the imaging chamber was functionalized with 1:1 molar ratio of bovine serum albumin (BSA) and BSA-biotin (1 mg mL$^{-1}$, Thermo Fisher Scientific), and then with Neutravidin (1 mg mL$^{-1}$, Sigma Aldrich), to tether the biotinylated liposomes. About 7 µL of the liposome solution was carefully pipetted (with a cut tip) into the imaging chamber and supplemented with RQ1 DNase (0.07 U µL$^{-1}$) to preclude gene expression outside liposomes. The chamber was sealed by sticking a 20 × 20 mm coverslip with a double-sided adhesive silicone sheet. Expression was performed directly on the confocal microscope at 37 °C for 1.5–6 h.

**Confocal microscopy**. A Nikon A1R Laser scanning confocal microscope equipped with an SR Apo TIRF ×100 oil-immersion objective was used to image liposomes and SLB in Min-FtsZ combined experiments and in in situ MinD/E expression kinetics. The 561 nm and 488 nm laser lines were used in combination with appropriate emission filters to image the Texas Red membrane dye and eGFP-MinC, respectively. The software NIS (Nikon) was used for image acquisition and identical settings were used for all experiments. Samples were mounted on a temperature-controlled stage maintained at 37 °C during imaging.

**Image analysis**. Image analysis was performed using custom scripts in MatLab version 2018a. Liposome images in the Texas Red channel were background corrected by morphological opening with a disk radius of four pixels. To enhance coherence of the liposome membrane signal, a series of rotated elongated Laplacian of Gaussian filters of size 10 pixels with SDs of 3 and 0.1 pixels was applied. The maximum projection of these filtered images was used as input for a seeded watershed segmentation, where the seed was derived from the segmentation of the previous image by image erosion with a disk radius dependent on the liposome size. For the first frame of the time-lapsed images, a user-supplied seed was used. A Gaussian filter of size 10 pixels with a sigma of 1.2 pixels was applied to all GFP images and the pixel intensities corresponding to the segmentation result were extracted to generate kymographs. As the length of the circumference changes during the time-series images when liposomes undergo shape changes, the individual lines of the kymograph were all adjusted to the number of pixels of the longest circumference using linear interpolation. Liposome elongation was calculated as the ratio of major to minor axis length of the ellipse that has the same normalized second central moments as the segmented liposome. The normalized perimeter length is the circumference of a circle with the same area, $A$, as the liposome and was calculated as $\pi \times \sqrt{4A/\pi}$.

The frequency of oscillations of the standing wave patterns on SLB was determined by calculating the autocorrelation function over time for each pixel and extracting the first peak of the function if existing. The median of those values for one image is reported. The wavelength and velocity of the traveling waves on SLB were analyzed by generating a kymograph along a line in the direction of the traveling wave. For the experiments combining MinDE with FtsA and FtsZ-Alexa647, images were preprocessed with an 8-pixel radius Gaussian blur filter before generating the kymographs. After binarization of the kymograph by Sobel edge detection, lines were detected using a linear Hough transform. The slope of the lines corresponds to the wave velocity, the spacing between the lines to the wavelength.

**Reporting summary**. Further information on research design is available in the Nature Research Reporting Summary linked to this article.

## Data availability

Data supporting the findings of this manuscript are available from the corresponding author upon reasonable request. A reporting summary for this Article is available as a Supplementary Information file. The source data underlying Fig. 1b–d, and Supplementary Figs. 1, 3, and 4 are provided as a Source Data file. This includes uncropped gels and raw data for the mass spectrometry analysis. Proteomic data have been uploaded on Panorama Public and can be accessed with the following URL (https://panoramaweb.org/Ye4TGO.url) and ProteomeXchange ID (PXD015686).

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

## Acknowledgements

We thank Esengül Yildirim for cloning the *minD* and *minE* genes, Ilja Westerlaken for purifying the eGFP-MinD and eGFP-MinC proteins, Jeremie Capoulade for assistance with fluorescence microscopy, Duco Blanken for fitting the mass spectrometry kinetics and for assistance with confocal microscopy, the group of Cees Dekker for providing us with the purified MinE, the group of Petra Schwille for providing the eGFP-MinD and eGFP-MinC coding plasmids, and the group of Germán Rivas for providing us with the purified FtsZ-Alexa647. Microscopy measurements were performed at the Kavli Nanolab Imaging Center Delft. This work was financially supported by the Netherlands Organization for Scientific Research (NWO/OCW) through the "NanoFront—Frontiers of Nanoscience" Gravitation grant, the "BaSyC – Building a Synthetic Cell" Gravitation grant (024.003.019), and the FOM program number 151.

## Author contributions

C.D. conceived and supervised the research. E.G., J.N., and C.D. designed the SLB and liposome experiments. D.F. and E.G. designed the LC-MS experiments. E.G. performed the SLB and liposome experiments. D.F. performed and analyzed the LC-MS experiments. J.N. and C.C. contributed preliminary bulk and SLB experiments. A. D. and C.F. analyzed the imaging data. C.D. and E.G. wrote the paper with inputs from all coauthors.

## Competing interests

The authors declare no competing interests.
