## [Peer Review File · Nature Communications]

Reviewers' Comments:

Reviewer #1:

Remarks to the Author:

The manuscript by Godino et al. reconstituted Min dynamics on both supported lipid bilayers and in encapsulated vesicles using in vitro synthesized Min proteins. Importantly, oscillation of Min proteins inside a liposome is coupled to membrane deformation. The observed Min dynamics encapsulated in liposome and subsequent membrane deformation was recently described by Schwille and co-worker, and the present work reported similar behaviors. The main advance of the work is the use of cell-free expressed Min proteins encoded in DNA template using the PURE system. This feature is important to have hereditary information in a synthetic cell such that when a synthetic cell divides, subsequent processes, including division, can take place again. This is challenging given the current state of research in this area, so the work is limited to demonstrating cell-free expressed Min dynamics that have been observed. In this regard, the biophysics of Min dynamics is not advanced. The manuscript is overall well written and well organized. However, there are numerous methodological questions that, in my opinion, are important that have not been addressed. Given this is the first work that uses cell-free expressed Min, a more thorough comparison between the present study and previous studies is necessary.

1. One of the deficiencies of this work is on reporting the final concentrations of cell-free synthesized proteins in the different assays. I appreciate the authors using quantitative mass spec to determine the bulk expression of the proteins which resulted in 15 and $\sim 3 \mu\text{M}$ of Min D and E. In Figure 1c for MinD, is there a reason why there is a drop-off at amino acid 230? These concentrations are much higher than typical concentrations used for reconstituting Min dynamics (typically at 1 to 3 μM of equimolar of MinD and E). In cells, MinD and E are roughly the same concentration. There is no discussion at any levels on how this changes the Min dynamics. Are these also the concentrations that the authors expected on the bilayer and when encapsulated? If not, the estimated concentrations in each assay need to be stated (were there dilution from other added components?). The authors mentioned something about interference with lipid film swelling and crowding during liposome formation that alters concentrations. This remark, which I don't fully understand, make it even more important to determine (or estimate) the concentrations MinD/E encapsulated in the liposomes.

2. Related to the above is that concentrations of proteins would depend on the time of expression. For the quantitation in Figure 1, a 3-hour expression is used. In other places in the manuscript, the authors have noted 1.5 h of expression and in Figure 3, 1-6 hour (method stated 1.5-6 hr) was stated. Presumably, the time points at which dynamics were observed would also be related to the concentrations of proteins at the instant in time. The kinetics of protein synthesis can probably be readily obtained using a plate reader for bulk reactions. Is this something that the authors have done? I see from Figure 3b that different dynamic behaviors were observed at the same time so it may not be possible to attribute the different dynamics to different MinD/E concentrations. Nevertheless, this point needs to be explicitly discussed.

3. It is puzzling to me why the authors chose to use purified eGFP-MinC for reporting MinDE dynamics in liposome. The rationale provided did not make sense. The bilayer experiments used eGFP-MinD, and then the authors switched to eGFP-MinC for the liposome encapsulation. The issue here is that MinC actually can inhibit pattern formation at high concentration (probably not as bad of an issue here since the authors do see dynamics). MinE needs to displace MinC from MinD in order to trigger ATP hydrolysis. Zieske and Schwille (eLife 2014) showed that high concentration of MinC inhibits patterns, and Loose et al (NSMB 2011) showed a low concentration of MinC increases wavelength and decreased velocity of Min dynamics. As stated earlier, most Min reconstitution use far lower concentration of MinD and MinE and generally do not use MinC. In the present case, MinC was used at 1 μM (there is discrepancy in the method where it states 2 μM). In cells, MinC is about 5% of that of MinD and MinE in copy number. It remains puzzling why the authors did not use labeled MinD as they did for the bilayer experiment. It would seem like the authors have tried this. I think it would be important to reconstitute this in the absence of MinC and see if similar behaviors in the liposome can be observed.

4. Curiously, if the authors include MinC, it would seem logical to also include FtsZ in the liposome as well. Since Min dynamics and membrane deformation have been reported in liposomes, showing different FtsZ pattern along with Min dynamics in liposome would be a novel biological aspect of the work.

5. The manuscript would benefit from more quantification of the Min dynamics, both in terms of supported membrane assay and in liposomes. There can also be more discussion comparing assays using cell-free expressed proteins vs. purified proteins. For instance, the wavelength from Figure 1 looks quite small, and this is expected given the high concentrations of the Min proteins. Have the authors quantified wave velocity and wavelengths for supported bilayer cases? I can imagine using purified MinE (for example) and cell-free expressed MinD to characterize Min wave and compare it to purified proteins. The wave characteristics can then be used to back out protein concentrations. Here, at the minimum, the authors should discuss the biophysics of Min waves between the two systems. I think it'd also be important to point out that as a cell-free expression system that Min proteins are produced with time (related to point 2 above), and could lead to changing Min patterns. Is this something that the authors have observed? I think that could be an interesting feature that would be distinct from using purified proteins.

Minor suggestions/comments:

- Introduction. At the end of first paragraph, 'membrane-forming' subsystem seems odd to me, may want to revise. At the end of third paragraph, large scale membrane remodeling, considering changing remodeling to deformation. Last paragraph of the introduction, encapsulation inside cell-sized liposomes was 'straightforward'. This does not seem like a good description encapsulation is generally challenging to many groups and stating it as straightforward somehow diminishes the work itself.
- Results and Discussion, first section. Using purified proteins of 'known' concentrations.
- Figure 1b. Could you put molecular weight markers for the gel?
- Results and Discussion, second section. The sentence 'Second, although the minE gene is directly.....was necessary' is a bit clunky and may require revision. It will be helpful to indicate that DnaK mix is the chaperone for readers that are not familiar with it. The way this is written requires some inference. On this note, what is the 'mix'? Is there any additional components besides DnaK? It'd be good to include the information in Figure 2a in the text, for instance the time of expression.
- Results and Discussion, third section. '...the advantage of not containing solvent'. Presumably, the authors meant organic solvent as water is a solvent. The authors may want to change the order of information describing Figure 3e to come before Figure 4.
- Figure 3e. Can the authors make the markers a bit larger and have the legend for the marker on the figure? For the figure here and later that displays fluorescence of eGFP-MinC on the membrane and in the lumen, I find the choice of color to be confusing since later in Figure 5, the fluorescence micrographs have magenta as the membrane dye signal. The authors may want to use a different color to denote fluorescence in the lumen.
- Figure 5. Figure 5a. Is there any reasons why the eGFP-MinC is so prominent external to the liposome in this example? This looks much more different compared to the other examples shown. Figure 5b. eGFP-MinC at the membrane is also much less discernible compared to other cases. Is there a reason why this is the case?
- Conclusion. The last statement on selecting liposomes with a desirable phenotype for further rounds of directed evolution sounds intriguing. However, if the dynamic phenotype is due to concentrations of proteins expressed, how this information will be carried forward to the next generation is not so clear. The authors are encouraged to describe this more clearly for the readers.
- Methods. Quantitative proteomics. What does 'fitted with 'instrumental' weighting' mean? Please elaborate this more in the text. Activity assays on supported membrane. The authors can also include how long is the cell-free expression reaction. Preparation of lipid-coated beads. What is the purpose of rhamnose?

Reviewer #2:

Remarks to the Author:

The present study is on reconstitution of Min proteins system by synthesizing both MinD and MinE in the liposome using purified cell-free expression system. Though many similar reconstitution study using purified proteins has been published by other groups (ref. 18-21), the study by simultaneous expression of Min proteins are novel and worth publishing. However, I think that the current study lacks biologically novel findings that may not be enough to attract broad communities. Thus, I recommend the authors to submit the manuscript to more specialized journals. Detailed comments are attached below.

1. P. 6, What is DnaK mix? What does it include? What is its effect?
2. P. 7, Is there any rational reason about trying only two conditions for lipid composition? How can it compared to other studies?
3. P. 7, Why didn't they try MinC expression?
4. P. 9, How do the current observations on oscillation features differ from (or similar with) other studies?
5. P. 9, How can it compared with other studies about the finding that Min self-organization is robust to the surface charge density?
6. P. 11, The finding that liposome can be deformed even with the absence of osmotic pressure may be biologically novel. The manuscript could be more significant to broad fields if detailed mechanisms of the finding are elucidated.

Reviewer #3:

Remarks to the Author:

This paper represents a proof of concept paper. The Min system is well characterized in vivo and in vitro – in fact it is the favorite system for reconstitution for pattern forming proteins. The long term goal is bottom up biology to make a cell. The immediate goal here is to reconstitute the Min oscillation in vesicles from PCR fragments, i.e. to express genes within vesicles whose products are functional. The MinD and MinE gene fragments are mixed with a commercial translation system, spiked with a GFP-tagged protein (MinC or MinD), encapsulated in lipid vesicles and observed by fluorescence microscopy. To observe dynamic movement of the proteins it was necessary to add ATP (over and above what is in the commercial system) and the DnaK chaperone during encapsulation. Also, to get suitable expression of MinE the gene construct has to be altered to remove some RNA secondary structure and optimize codon usage. Once this was done various patterns were observed that resembled that reported in ref 21.

Some comments:

The paper represents a tour de force in the sense of getting the system operational. A lot of work went into getting the kinks out of the system – getting expression working as well as adding a chaperone.

The commercial system for protein production must be quite inefficient due to the length of time of incubation (generally many hours, 3 or greater). How reproducible is the system? Does the reconstitution work every time?

Did the authors consider using a PCR fragment containing MinD and MinE in tandem like they exist on the chromosome – they appear to be translationally coupled and most in vivo systems have them in tandem. Then a control could be with a stop codon in MinE so only MinD would be expressed. How much effort went into varying the ratios of the PCR fragments to get oscillation?

Why do the authors use mass spec to determine the concentration of MinD and MinE – on the

manufacturers website they run SDS-PAGE gels and stain. One could just run a known amount of MinD and MinE alongside the in vitro reactions.

Does tethering the liposomes to the support have any effect on their behavior?

Reviewer #4:

Remarks to the Author:

The article by Godin et al describes engineering a self-organizing biochemical network model using Min family proteins in cell-free protein expression system. This is very interesting work, well thought out, and the experimental evidence convincingly and sufficiently supports the conclusions drawn in the paper.

The rationale of this project is very clear to someone who is involved in this field. This work attempts to address one of the biggest unsolved problems towards the construction of artificial living cell from non-living and engineerable components. This is, however, not sufficiently clearly communicated in the introduction, abstract, or anywhere in the paper. The authors seem to take for granted introductory level of understanding of synthetic cell field, I think for a general readership journal like Nature Communications this rationale should be made much clearer.

Similarly, the choice of the particular type of cell-free protein expression system is not explained. One familiar with the field can understand why E coli cell-free protein system was chosen, but that was never explained sufficiently in the text.

Along the same lines, why was PURE system used, instead of whole cell protein expression extract? If this work is to be used as a stepping stone towards the construction of synthetic cell, yields and efficiency of whole cell extract are much higher, offering potentially higher chance of success, than PURE.

In figure 1, two biological replicates were reported. I was always under impression that minimum three independent replicates are necessary to have any confidence in the results. Is the standard deviation reported on figure 1c taken from technical replicates? Error bars should be reported using actual independent replicates, not technical replicates. Unless author's definition of technical and biological replicate is different from my understanding of it (biological replicate would be with different batches of enzymes, technical is set up from the same master mix) there should be minimum one more true replicate.

Why was Pc/Pg liposome system used? Authors report testing at least one more composition, but only give rationale for the PC/PG/PE/CL as being similar to E coli membrane. It would be useful to give explanation why, choosing simplified membrane system, PC/PG was used and not any other possibility, and why the particular molar ratio of those lipid were used.

Has it ever been tested that activity of eGFP-MinD and eGFP-MinC is not altered by the large, hydrophilic GFP fusion? It is clear the proteins have some very clear effect, as shown on Figure 5 and in supplementary videos, but it would be worth discussing if this is comparable to expected endogenous activity.

Figure 4 is unclear. The panels a – c are difficult to understand. Some more description, perhaps labeling of which pattern of transition we're seeing, would make it much clearer.

I am curious if authors ever tried co-expression, or tests of activity, together with FtsZ? The FtsZ was successfully expressed in cell-free systems, it would make this work much stronger if complex of all three proteins was tested. I understand if this is outside of the scope of this work though.

Rebuttal letter for:

Manuscript: "De novo synthesized Min proteins oscillate in liposomes and drive membrane deformation" by Godino et al.

Article reference: NCOMMS-19-14437-T

We thank the reviewers for finding our work of 'considerable potential interest' and for their constructive comments that helped us improve the manuscript. The Referee reports are in *blue text* and our point-by-point responses are in *black text*. Changes in the manuscript (main text and Supplementary information) are highlighted in *red text*.

We are pleased to provide you with a substantially revised manuscript that addresses all referees' concerns. In particular, we included:

- a) A more advanced quantitative analysis of the Min system and a better comparison to the conventionally reconstituted Min system behaviour,
- b) New experiments with the synthesized MinC protein and integration of FtsZ,
- c) Additional mass spectrometry experiments that completely address the reproducibility concerns.
- d) A clearer description of the innovative aspects of our work.

We would like to mention that we initiated this study four years ago and performed all experiments independently of the work of Schwille and coworkers (Ref.21).

Reviewer #1 (Remarks to the Author):

The manuscript by Godino et al. reconstituted Min dynamics on both supported lipid bilayers and in encapsulated vesicles using in vitro synthesized Min proteins. Importantly, oscillation of Min proteins inside a liposome is coupled to membrane deformation. The observed Min dynamics encapsulated in liposome and subsequent membrane deformation was recently described by Schwille and co-worker, and the present work reported similar behaviors. The main advance of the work is the use of cell-free expressed Min proteins encoded in DNA template using the PURE system. This feature is important to have hereditary information in a synthetic cell such that when a synthetic cell divides, subsequent processes, including division, can take place again. This is challenging given the current state of research in this area, so the work is limited to demonstrating cell-free expressed Min dynamics that have been observed. In this regard, the biophysics of Min dynamics is not advanced. The manuscript is overall well written and well organized. However, there are numerous methodological

questions that, in my opinion, are important that have not been addressed. Given this is the first work that uses cell-free expressed Min, a more thorough comparison between the present study and previous studies is necessary.

Reply: We thank the referee for his/her careful review and very constructive comments. Indeed, the main challenge that we tackled in the present study is to establish working methodologies for cell-free expression of active Min proteins on planar membranes and within liposomes. As developed below, we conducted a more quantitative analysis of the Min dynamics, including with purified proteins. In addition, as elaborated under point 4, we succeeded in combining the MinCDE and FtsA-FtsZ systems, which, to the best of our knowledge, has never been reported, and therefore contributes to the scientific advance of the biology of these two systems.

1. One of the deficiencies of this work is on reporting the final concentrations of cell-free synthesized proteins in the different assays. I appreciate the authors using quantitative mass spec to determine the bulk expression of the proteins which resulted in 15 and ~ 3 μM of Min D and E. In Figure 1c for MinD, is there a reason why there is a drop-off at amino acid 230?

Reply: We have performed quantitative mass spectrometry over more than 30 different proteins expressed in the PURE system (unpublished yet) and the lower abundance of C-terminal peptides, as observed with MinD, is a very common pattern. Ribosome processivity error is the most probable cause, but the exact mechanism remains to be elucidated. We added on page 16, lines 20-21, a sentence referring to ribosome processivity error and linked this shortcoming of the PURE system to the challenge of expressing all five Min and FtsZ proteins (see point 4).

These concentrations are much higher than typical concentrations used for reconstituting Min dynamics (typically at 1 to 3 μM of equimolar of MinD and E). In cells, MinD and E are roughly the same concentration. There is no discussion at any levels on how this changes the Min dynamics.

Reply: Concentrations of ~ 15 μM and ~ 3 μM of MinD and MinE, respectively, are typically obtained after 3 h expression in test tube reactions. As we will further comment under point 5, we conducted new experiments of MinDE expression directly on top of an SLB and recorded the Min dynamics at different time points (over several hours) during which protein concentrations increase (new Fig. 2d). Interestingly, although protein concentrations vary from ~ 4 μM to ~ 20 μM for MinD, and from ~ 2 μM to ~ 5 μM for MinE between 1 h and 4 h, no marked changes of the Min dynamic parameters were observed (new Supplementary Fig. 6), indicating robustness of the Min behaviour across this concentration range. Actually the MinE to MinD ratio varies less than the absolute concentrations (new Supplementary Fig. 5), which may explain the overall steady behaviour. Moreover, we performed additional experiments with 1 μM of purified MinD and 1 μM of purified MinE, and found similar dynamic parameters (new Supplementary Fig. 7 and new Movie 8).

Are these also the concentrations that the authors expected on the bilayer and when encapsulated? If not, the estimated concentrations in each assay need to be stated (were there dilution from other added components?).

Reply: We do expect similar concentrations on the SLB since the expression conditions (DNA concentration, temperature, PURE system composition) were identical as in test tube reactions. In the revised manuscript, we specified the concentrations of synthesized Min proteins in SLB assays when presenting the results of the new Fig. 2d.

In liposomes, the situation is completely different. Due to the stochasticity of component encapsulation during liposome formation, we expect every liposome to enclose a different molecular composition. This heterogeneity leads to a broad range of protein synthesis yield, even in liposomes of the same size and at a given time point. These effects have been fully quantified in a recent article from our group (Ref.24 in the original manuscript). This point is mentioned on page 12, lines 4-8.

The dilution effects are minor in all SLB experiments, except in the assays with FtsA-FtsZ, where each of the expressed protein samples are diluted by a factor around three (each 5.5 μ L in total 20 μ L).

The authors mentioned something about interference with lipid film swelling and crowding during liposome formation that alters concentrations. This remark, which I don't fully understand, make it even more important to determine (or estimate) the concentrations MinD/E encapsulated in the liposomes.

Reply: In the light of the newly performed experiments with eGFP-MinD, we decided to modify the corresponding sentence on page 9, line 17.

As discussed above and demonstrated in Ref.24 with the expression of the fluorescent protein YFP, the concentrations of MinD/E can largely differ between liposomes. Whereas the concentration of YFP could readily be assessed thanks to its autofluorescence, determining the levels of synthesized MinD/E would be very challenging. We considered various options, such as incorporation of fluorescent unnatural amino acids, FIAsh and ReAsH labelling of tetracysteine-modified proteins, single-liposome mass spectrometry. However, all the envisaged methods suffer from their own caveats and would not allow accurate quantification of active proteins. Importantly, the poor sensitivity of the MinDE dynamics to the protein concentrations in SLB assays suggests that Min oscillations in liposomes might occur for a broad range of MinD/E concentrations.

2. Related to the above is that concentrations of proteins would depend on the time of expression. For the quantitation in Figure 1, a 3-hour expression is used. In other places in the manuscript, the authors have noted 1.5 h of expression and in Figure 3, 1-6 hour (method stated 1.5-6 hr) was stated. Presumably, the time points at which dynamics were observed would also be related to the concentrations of proteins at the instant in time. The kinetics of protein synthesis can probably be readily obtained using a plate reader for bulk reactions. Is this something that the authors have done?

Reply: We clarified the expression duration in all assays.

We have not considered spectrofluorometry or absorbance measurements (plate reader) to characterize the kinetics of protein synthesis because this would require to label MinD/E and, in most instances, to introduce a peptide tag, which might affect transcription/translation. Alternatively, we measured the kinetics of MinD/E concentrations by carrying out quantitative mass spectrometry at different points of test tube reactions. The results are reported in the new Fig. 1d and the kinetic parameters are indicated on page 5, lines 2-4.

I see from Figure 3b that different dynamic behaviors were observed at the same time so it may not be possible to attribute the different dynamics to different MinD/E concentrations. Nevertheless, this point needs to be explicitly discussed.

Reply: As discussed above, liposomes may contain different concentrations of synthesized Min proteins at a given time point. Therefore, one cannot exclude the possibility that different dynamic behaviors arise at different MinD/E concentrations.

3. It is puzzling to me why the authors chose to use purified eGFP-MinC for reporting MinDE dynamics in liposome. The rationale provided did not make sense. The bilayer experiments used eGFP-MinD, and then the authors switched to eGFP-MinC for the liposome encapsulation.

Reply: The line of reasoning for switching the fluorescent reporter in liposome assays was the following. First, the presence of membrane binding proteins, like MinD but not MinC, in the lipid film swelling medium can reduce the efficiency of liposome formation and encapsulation by interacting to the membrane. Second, protein encapsulation is a random process that leads to a heterogeneity of the internal concentration of proteins between liposomes. In some rare cases, the amount of encapsulated compound can largely exceed theoretical predications with a vesicle partitioning that rather obeys a power law than a Poisson distribution (see for instance the work of Luisi and coworkers). Therefore, even when using a low bulk concentration of eGFP-MinD (here 100 nM), there exists a chance for a small subset of liposomes to contain a higher concentration, potentially contributing to generate the oscillations. To rule out this possibility, we thus opted for eGFP-MinC. Moreover, on page 9, line 18, we added that purified eGFP-MinD could bind to the external surface of the liposome membrane, which could hamper easy visualization of the Min oscillation on the inner surface.

Even if MinC is known to modulate the Min dynamics, it does not produce the waves. Hence, any detected Min oscillations can unambiguously be attributed to synthesized MinD/E. Another motivation is related to the eGFP signal intensity. Using eGFP-MinD as a reporter implies to supply it at a low concentration to not drive wave formation. However, at such low concentrations time-lapse confocal imaging becomes challenging, as discussed below. In contrast, eGFP-MinC can be supplemented at higher concentration enabling higher resolution imaging. For these reasons, we initially opted for eGFP-MinC as a superior fluorescent reporter in liposome assays. On SLB, the experimental constrains are different and eGFP-MinD at low concentration appeared to be a good choice.

We performed additional control experiments using purified eGFP-MinD in place of purified eGFP-MinC inside liposomes. Similar dynamic behaviors were observed, though the imaging quality was not as good as with eGFP-MinC (new Supplementary Fig. 11).

The issue here is that MinC actually can inhibit pattern formation at high concentration (probably not as bad of an issue here since the authors do see dynamics). MinE needs to displace MinC from MinD in order to trigger ATP hydrolysis. Zieske and Schwille (eLife 2014) showed that high concentration of MinC inhibits patterns, and Loose et al (NSMB 2011) showed a low concentration of MinC increases wavelength and decreased velocity of Min dynamics.

Reply: We are familiar with these studies, the latter being cited in the manuscript. Indeed, we first tried with a relatively high concentration of eGFP-MinC and it worked. We added a discussion on this point on page 13, lines 4-11, and performed additional with a lower concentration of eGFP-MinC (Supplementary Fig. 11).

As stated earlier, most Min reconstitution use far lower concentration of MinD and MinE and generally do not use MinC. In the present case, MinC was used at 1 μM (there is discrepancy in the method where it states 2 μM). In cells, MinC is about 5% of that of MinD and MinE in copy number. It remains puzzling why the authors did not use labeled MinD as they did for the bilayer experiment. It would seem like the authors have tried this. I think it would be important to reconstitute this in the absence of MinC and see if similar behaviors in the liposome can be observed.

Reply: Thank you for pointing out this mistake, we fixed it.

No, we had not tried to use labelled MinD in liposomes. We performed new experiments and reconstituted Min oscillations in the absence of MinC (with supplied 200 nM eGFP-MinD) and at lower concentration of MinC (0.4 μM instead of 1 μM). The results are reported in Supplementary Fig. 11 and discussed as indicated above. As expected, low concentrations of either eGFP-MinD or eGFP-MinC enabled Min dynamics (pulsing behaviour), though the lower signal intensity made long time-lapse imaging challenging. In cells, MinC concentration is low relative to that of MinD/E because it acts as a regulator of FtsZ polymerization. In liposomes, we found that a relatively high concentration of eGFP-MinC offers the best trade-off to visualize Min oscillations.

4. Curiously, if the authors include MinC, it would seem logical to also include FtsZ in the liposome as well. Since Min dynamics and membrane deformation have been reported in liposomes, showing different FtsZ pattern along with Min dynamics in liposome would be a novel biological aspect of the work.

Reply: As suggested, we performed additional SLB experiments and demonstrated dynamic FtsZ patterns along with planar Min waves. We dedicated a new section in Results and Discussion on the integration of the Min and FtsZ systems and a new Fig. 6 (with associated Supplementary information figures). These experiments involved the cell-free expression of two new proteins, MinC and FtsA, the latter serving as the membrane anchor of FtsZ. A new protein gel showing the expression of MinC is depicted in the new Fig. 1b. To our knowledge, this result is the first demonstration that FtsA-FtsZ co-filaments can form dynamic patterns when coupled to MinCDE. Of note, this assay relies on the expression of four different proteins, FtsA and MinCDE, while the purified labelled FtsZ-Alexa647 was used. Liposome experiments will undoubtedly be extremely challenging as all four proteins will have to be co-synthesized within individual vesicles. Future investigations will be performed to tackle the current limitations of the PURE system to express multiple proteins at micromolar concentrations each. We discussed this point on page 16.

5. The manuscript would benefit from more quantification of the Min dynamics, both in terms of supported membrane assay and in liposomes. There can also be more discussion comparing assays using cell-free expressed proteins vs. purified proteins. For instance, the wavelength from Figure 1

looks quite small, and this is expected given the high concentrations of the Min proteins. Have the authors quantified wave velocity and wavelengths for supported bilayer cases?

Reply: In Fig. 3e, we already quantified the Min pulsing periodicity in liposomes of different sizes and lipid composition. As recommended, we quantified the wave velocity and wavelength in SLB assays. The results are reported in Supplementary Fig. 6 and in the main text on page 8. We also compared the Min dynamic parameters with literature values obtained with purified proteins, and performed new experiments with purified proteins in the PURE system. We noticed that the wavelength from Fig. 2b (we guess the reviewer is referring to that figure) is quite small. Actually, it is not representative of the many patterns we have imaged. We chose this figure because it clearly displays the most frequent dynamic behaviours. In the revised manuscript, more images of Min patterns are shown in new Fig. 2b,c.

I can imagine using purified MinE (for example) and cell-free expressed MinD to characterize Min wave and compare it to purified proteins. The wave characteristics can then be used to back out protein concentrations. Here, at the minimum, the authors should discuss the biophysics of Min waves between the two systems.

Reply: We performed additional experiments with purified MinE/D as explained above. The biophysics of the different Min waves are very similar. This was expected given the robustness of the patterns with respect to MinD/E concentrations, at least within the range of $>2 \mu\text{M}$ proteins used. This lack of sensitivity over protein concentration makes it difficult to accurately assess the concentration of expressed proteins in a purified/expressed assay configuration.

I think it'd also be important to point out that as a cell-free expression system that Min proteins are produced with time (related to point 2 above), and could lead to changing Min patterns. Is this something that the authors have observed? I think that could be an interesting feature that would be distinct from using purified proteins.

Reply: This is an excellent suggestion that we have attempted to explore. The corresponding experimental design and results are reported in the new Fig. 2c and extensively discussed. In situ expression of MinD/E on top an SLB allowed us to monitor Min dynamics over several hours concurrently to protein production. This unique setup enables to observe in real-time the evolution of Min pattern at increasing concentrations of MinD/E, something that has not been achieved using purified proteins.

Minor suggestions/comments:

- Introduction. At the end of first paragraph, 'membrane-forming' subsystem seems odd to me, may want to revise.

Reply: We changed it into 'compartment'.

At the end of third paragraph, large scale membrane remodeling, considering changing remodeling to deformation.

Reply: We changed it as suggested.

Last paragraph of the introduction, encapsulation inside cell-sized liposomes was 'straightforward'. This does not seem like a good description encapsulation is generally challenging to many groups and stating it as straightforward somehow diminishes the work itself.

Reply: We replaced 'straightforward' by 'successfully realized'.

- Results and Discussion, first section. Using purified proteins of 'known' concentrations.

Reply: Thanks, we corrected it.

- Figure 1b. Could you put molecular weight markers for the gel?

Reply: We did.

- Results and Discussion, second section. The sentence 'Second, although the minE gene is directly.....was necessary' is a bit clunky and may require revision. It will be helpful to indicate that DnaK mix is the chaperone for readers that are not familiar with it. The way this is written requires some inference. On this note, what is the 'mix'? Is there any additional components besides DnaK? It'd be good to include the information in Figure 2a in the text, for instance the time of expression.

Reply: We added 'a cocktail of highly purified chaperone proteins' on page 6, line 25. In the Methods section, we provided a description of the DnaK mix as described by the supplier.

The time of expression (3 h) is already indicated in Fig.2a.

- Results and Discussion, third section. '....the advantage of not containing solvent'. Presumably, the authors meant organic solvent as water is a solvent.

Reply: Indeed, we added 'organic'.

The authors may want to change the order of information describing Figure 3e to come before Figure 4.

Reply: We followed the suggestion.

- Figure 3e. Can the authors make the markers a bit larger and have the legend for the marker on the figure?

Reply: We changed it as suggested.

For the figure here and later that displays fluorescence of eGFP-MinC on the membrane and in the lumen, I find the choice of color to be confusing since later in Figure 5, the fluorescence micrographs have magenta as the membrane dye signal. The authors may want to use a different color to denote fluorescence in the lumen.

Reply: We switched the colors for the lumen and cortex signals.

- Figure 5. Figure 5a. Is there any reasons why the eGFP-MinC is so prominent external to the liposome in this example? This looks much more different compared to the other examples shown.

Reply: In Fig. 5a we used different settings to give the best contrast between fluorescent and nonfluorescent lumen. In Fig. 5c, the external medium is less fluorescent due to dilution with sucrose (and the settings were lower).

Figure 5b. eGFP-MinC at the membrane is also much less discernible compared to other cases. Is there a reason why this is the case?

Reply: This is true. One possible explanation is that eGFP-MinC is not evenly distributed at the membrane but preferentially accumulates at areas on the membrane that are out of focus. Depletion of eGFP-MinC from the lumen will lead to a lower signal (as clearly observed) even if membrane recruitment occurs on a different imaging plane.

- Conclusion. The last statement on selecting liposomes with a desirable phenotype for further rounds of directed evolution sounds intriguing. However, if the dynamic phenotype is due to concentrations of proteins expressed, how this information will be carried forward to the next generation is not so clear. The authors are encouraged to describe this more clearly for the readers.

Reply: We clarified by specifying that the DNA variant of the selected liposomes will be isolated and amplified.

- Methods. Quantitative proteomics. What does 'fitted with 'instrumental' weighting' mean? Please elaborate this more in the text.

Reply: We clarified.

Activity assays on supported membrane. The authors can also include how long is the cell-free expression reaction.

Reply: We've been careful to consistently include the expression time in the revised version.

Preparation of lipid-coated beads. What is the purpose of rhamnose?

Reply: It enhances lipid film swelling. It is included in an existing protocol that is cited in this section.

Reviewer #2 (Remarks to the Author):

The present study is on reconstitution of Min proteins system by synthesizing both MinD and MinE in the liposome using purified cell-free expression system. Though many similar reconstitution study

using purified proteins has been published by other groups (ref. 18-21), the study by simultaneous expression of Min proteins are novel and worth publishing. However, I think that the current study lacks biologically novel findings that may not be enough to attract broad communities. Thus, I recommend the authors to submit the manuscript to more specialized journals. Detailed comments are attached below.

Reply: We believe that the findings that sequence optimization and DnaK chaperone favour the expression of active proteins contribute to the biological novelty from which the entire synthetic biology community will benefit. Moreover, we anticipate that the established protocol to encapsulate a synthetic Min system in liposome can readily be applied to confine other biological systems into cell-sized liposomes. The likely contribution of MinE to the liposome deformation mechanism is also new. As discussed below, we also demonstrated in the revised manuscript that MinC, along with FtsA and FtsZ can be integrated to the oscillating MinDE and generate dynamic patterns of FtsZ that have not been observed so far. Of note, this is the first time that the complete Min system, MinCDE, is reconstituted in liposomes. We believe that the innovative elements of our work are more clearly emphasized in the revised manuscript.

1. P. 6, What is DnaK mix? What does it include? What is its effect?

Reply: We clarified the role and composition of the DnaK mix on page 6 line 25, and in materials and methods.

2. P. 7, Is there any rational reason about trying only two conditions for lipid composition? How can it be compared to other studies?

Reply: The minimal PC/PG lipid composition is commonly used in SLB assays, including for reconstitution of the Min system (ref. 17, 19, 21). We included this information on page 9, line 12. It has the advantage to effectively produce stable planar membranes and it contains a physiological surface density of negative charges (through PG). The PC/PG/PE/CL composition resembles more closely the *E. coli* membrane composition and has proved to be compatible with in vitro gene expression (e.g. see 24). In most Min reconstitution assays, model membrane systems composed of the *E. coli* polar lipids have been used (ref. 16, 18, 20). Actually, the main factor for successful Min system reconstitution is the presence of negatively charged lipids (ref. 17), a requirement that is satisfied in both lipid compositions used in the present study.

3. P. 7, Why didn't they try MinC expression?

Reply: Our main goal was to demonstrate that the drivers of Min dynamics, MinD and MinE, can be functionally expressed on SLB and inside liposomes. We agree that a natural extension of this work includes the expression of MinC. In the revised manuscript, we designed, constructed and expressed in the PURE system a linear DNA coding for MinC (new Fig.1b, new Fig. 6 and corresponding Supporting information figures, new Supplementary Fig. 4). We also quantified the level of synthesized MinC by mass spectrometry (Supplementary Fig. 3 and 4). Finally, the full spectrum of MinC activity was confirmed by showing that the synthesized MinC can be recruited to the membrane and inhibit polymerization of FtsZ into filaments (new Fig. 6 and corresponding Supporting information figures).

When combining MinC, MinDE and FtsA-FtsZ, dynamic patterns of FtsZ were observed for the first time.

4. P. 9, How do the current observations on oscillation features differ from (or similar with) other studies?

Reply: We provided in the revised version a more thorough comparison of the Min dynamic parameters between purified and synthesized Min proteins. We also performed SLB assays with purified Min proteins in a PURE system background to ascertain that the experimental conditions were similar. The results, reported in Supplementary Fig. 7 and in the main text, show that the wavelengths and wave velocity are similar between expressed and purified proteins.

5. P. 9, How can it compared with other studies about the finding that Min self-organization is robust to the surface charge density?

Reply: In Ref. 17, the authors found that Min self-organization is robust over a wide range of anionic lipid densities (0 to 53%).

6. P. 11, The finding that liposome can be deformed even with the absence of osmotic pressure may be biologically novel. The manuscript could be more significant to broad fields if detailed mechanisms of the finding are elucidated.

Reply: Future investigations will concentrate on formulating a mechanistic model of liposome deformation. This will necessitate molecular dynamic simulations, theoretical analysis of membrane mechanics and further experiments. Our data underline the possible role of MinE in directly contributing to membrane deformation, a mechanism that has not been suggested in ref. 21.

We believe that our results are significant to the broader field of active soft matter. Hence, we decided to cite a relevant study (new ref. 37) and to emphasize the importance of our findings to that field.

Reviewer #3 (Remarks to the Author):

This paper represents a proof of concept paper. The Min system is well characterized in vivo and in vitro – in fact it is the favorite system for reconstitution for pattern forming proteins. The long term goal is bottom up biology to make a cell. The immediate goal here is to reconstitute the Min oscillation in vesicles from PCR fragments, i.e. to express genes within vesicles whose products are functional. The MinD and MinE gene fragments are mixed with a commercial translation system, spiked with a GFP-tagged protein (MinC or MinD), encapsulated in lipid vesicles and observed by fluorescence microscopy. To observe dynamic movement of the proteins it was necessary to add ATP (over and above what is in the commercial system) and the DnaK chaperone during encapsulation. Also, to get suitable expression of MinE the gene construct has to be altered to remove some RNA secondary structure and optimize codon usage. Once this was done various patterns were observed that resembled that reported in ref 21.

Some comments:

The paper represents a tour de force in the sense of getting the system operational. A lot of work went into getting the kinks out of the system – getting expression working as well as adding a chaperone.

Reply: We confirm that much effort has been spent to design the DNA constructs, optimize the gene expression conditions and establish protocols for imaging the Min dynamics in SLB and liposome assays. It is our hope that the developed methodologies will benefit other groups interested in cell-free protein synthesis, DNA-directed self-organization and reconstitution of biological functions.

The commercial system for protein production must be quite inefficient due to the length of time of incubation (generally many hours, 3 or greater). How reproducible is the system? Does the reconstitution work every time?

Reply: The PURE system suffers from some limitations (production period typically limited to <6 h, yield usually lower than with *E. coli* extracts). However, its performance is good enough to enable formation of planar Min waves already after 1 h expression (see new Fig. 2d), to express multiple proteins in a functional state, namely MinD, MinE, MinC and FtsA, the latter two being described in the revised manuscript. Moreover, MinD could be expressed at up to 25 μ M, which is quite high for batch mode reactions. Although MinD/E production is quite robust, the exact concentration of synthesized proteins can significantly vary from one reaction to another. This variability is illustrated in the new Fig. 1d, where three independent kinetics of MinD/E co-expression reactions are displayed. Supported bilayer assays are highly reproducible, presumably because of the robustness of the Min dynamics over MinD/E concentrations under the studied conditions. Success rate of the liposome experiments is more variable and we listed possible reasons on page 12.

Did the authors consider using a PCR fragment containing MinD and MinE in tandem like they exist on the chromosome – they appear to be translationally coupled and most in vivo systems have them in tandem. Then a control could be with a stop codon in MinE so only MinD would be expressed.

Reply: Expressing multiple genes in the form of an operon, i.e. one transcript encodes multiple proteins, is usually not recommended for cell-free protein synthesis. Protein yield is usually higher when every gene is expressed in the form of a transcriptional cassette. One reason is for instance that long mRNAs are more prone to fold into inhibitory secondary structures that impede translation initiation. A more detailed discussion, along with experimental data on RNA-based bottlenecks in cell-free gene expression, can be found in our recent article (cited as Ref. 23).

How much effort went into varying the ratios of the PCR fragments to get oscillation?

Reply: We initially tried with 5 nM of each MinD- and MinE-coding DNA and we failed reconstituting planar waves. We then decided to use a higher concentration of the *minE* gene compared to *minD* (8 nM and 4 nM, respectively) to compensate for the higher yield of synthesized MinD and produce more equimolar amounts of the two proteins. We obtained positive results and decided to keep this parameter constant for the next experiments.

Why do the authors use mass spec to determine the concentration of MinD and MinE – on the manufacturers website they run SDS-PAGE gels and stain. One could just run a known amount of MinD and MinE alongside the in vitro reactions.

Reply: The protein gels in Supplementary Fig. 1 show the multiple bands from PURE system proteins (negative control lanes). When the yield of synthesized protein is very high, i.e. largely exceeds the amount of PURE system proteins at the relevant molecular weight, then the expressed protein is visible on an SDS-PAGE gel. This is for instance the case for MinD, but not for MinE and MinC. A drawback of this method for quantitative purposes is that the size resolution is rather poor. In other words, if the expressed protein is truncated, the full-length and truncated translation products might be distinguishable on gel. In contrast, mass spectrometry allowed us to identify C-terminal peptides for accurate quantification of full-length proteins. In the case of MinD, one can note a drop of the amount of the most C-terminal peptide indicating premature translation arrest. The corresponding molecular mass difference will unlikely be visible on an SDS-PAGE gel.

Nonetheless, the suggested method is relevant, and we applied it to quantify the concentration of other proteins expressed in the PURE system (see Ref. 28).

Does tethering the liposomes to the support have any effect on their behavior?

Reply: We cannot exclude that physical attachment of liposomes to the glass surface may affect its deformability. Actually, not all liposomes are tethered. Some are floating and can hardly be imaged for a long period, while some vesicles are adhering to others. We believe that those liposomes that are not constrained to the support are more prone to deform.

Reviewer #4 (Remarks to the Author):

The article by Godin et al describes engineering a self-organizing biochemical network model using Min family proteins in cell-free protein expression system. This is very interesting work, well thought out, and the experimental evidence convincingly and sufficiently supports the conclusions drawn in the paper.

Reply: Thank you.

The rationale of this project is very clear to someone who is involved in this field. This work attempts to address one of the biggest unsolved problems towards the construction of artificial living cell from non-living and engineerable components. This is, however, not sufficiently clearly communicated in the introduction, abstract, or anywhere in the paper. The authors seem to take for granted introductory level of understanding of synthetic cell field, I think for a general readership journal like Nature Communications this rationale should be made much clearer.

Reply: As suggested, we added a sentence on page 2 line 9 and we elaborated on the significance of building an artificial cell for the general readership. in a new paragraph at the beginning of the Conclusion.

Similarly, the choice of the particular type of cell-free protein expression system is not explained. One familiar with the field can understand why E coli cell-free protein system was chosen, but that was never explained sufficiently in the text.

Along the same lines, why was PURE system used, instead of whole cell protein expression extract? If this work is to be used as a stepping stone towards the construction of synthetic cell, yields and efficiency of whole cell extract are much higher, offering potentially higher chance of success, than PURE.

Reply: We agree that this point requires more attention. On page 3 lines 19-22, we better motivated the use of PURE system over whole cell extracts. The choice of the type of PURE system is relevant for the more specialized readership and was already included in the original manuscript on page 4, lines 8-9.

In figure 1, two biological replicates were reported. I was always under impression that minimum three independent replicates are necessary to have any confidence in the results. Is the standard deviation reported on figure 1c taken from technical replicates? Error bars should be reported using actual independent replicates, not technical replicates. Unless author's definition of technical and biological replicate is different from my understanding of it (biological replicate would be with different batches of enzymes, technical is set up from the same master mix) there should be minimum one more true replicate.

Reply: We concur with the reviewer's definition of independent vs technical replicates, as well as with the necessity to report data from a minimum of three independent replicates (also called 'biological repeats' in the text). We had performed more measurements in slightly different conditions (e.g. single gene expression) but did not include them in the analysis. For the revised manuscript, we carried out three additional independent repeats (kinetics measurements), leading to a total of five biological repeats. We also included mass spectrometry quantification of expressed MinC, for which four independent experiments were conducted. For each biological repeat of the end-point measurements, three technical replicates were analysed. In new Fig. 1c, the average over all concentration values (independent plus technical replicates) and the individual data points for each biological repeat are displayed. In new Fig. 1d, three independent kinetic measurements are shown. The legend of Fig. 1 has been modified to include this information.

Why was Pc/Pg liposome system used? Authors report testing at least one more composition, but only give rationale for the PC/PG/PE/CL as being similar to E coli membrane. It would be useful to give explanation why, choosing simplified membrane system, PC/PG was used and not any other possibility, and why the particular molar ratio of those lipid were used.

Reply: The binary lipid composition PC/PG is commonly used in liposome research. We added this information on page 9, line 12. PC provides a stable lipid matrix while PG provides a negative surface charge that facilitates lipid film swelling and promotes interaction with membrane-associated proteins.

Has it ever been tested that activity of eGFP-MinD and eGFP-MinC is not altered by the large,

hydrophilic GFP fusion? It is clear the proteins have some very clear effect, as shown on Figure 5 and in supplementary videos, but it would be worth discussing if this is comparable to expected endogenous activity.

Reply: Both fusion proteins have successfully been used in Min reconstitution experiments. Noteworthy, eGFP-MinD is usually utilized in a mixture with unlabelled MinD (ratio labelled/unlabelled <3) because it can alter Min dynamics when present at high molar fraction [ref 20]. Mindful of this problem, we exclusively used eGFP-MinD at low concentration, way below the concentration of expressed unlabelled MinD. In typical SLB assays, the concentration of eGFP-MinD is 100 nM, while the estimated concentration of synthesized MinD is around 14 μ M. eGFP-MinC is less commonly used but has been shown to be active (ref 39). We have not performed a comparative activity analysis between eGFP-MinC and the expressed MinC.

Figure 4 is unclear. The panels a – c are difficult to understand. Some more description, perhaps labeling of which pattern of transition we're seeing, would make it much clearer.

Reply: To improve clarity, we appended a vertical line on the kymographs to separate the two behaviors and we added an explanatory sentence in the legend.

I am curious if authors ever tried co-expression, or tests of activity, together with FtsZ? The FtsZ was successfully expressed in cell-free systems, it would make this work much stronger if complex of all three proteins was tested. I understand if this is outside of the scope of this work though.

Reply: We agree that showing proper integration of the Min and FtsZ subsystems would be recognized as an important achievement. We first successfully expressed MinC and demonstrated its full activity spectrum regarding its traveling on the MinDE waves and inhibition of FtsZ polymerization. Then, we reconstituted the complete MinCDE-FtsA-FtsZ system and revealed the creation of dynamic patterns of FtsZ. We dedicated a new section in Results and Discussion and a new Fig. 6 (with associated Supplementary figures and movies), where we report the main results.

Reviewers' Comments:

Reviewer #1:

Remarks to the Author:

The revised manuscript has improved in several regards and the authors have taken care to address the main concerns that I raised previously. I was particularly impressed by the additional work that included FtsA and FtsZ in the present system, and I think this data adds significantly to the novelty of the science that was previously unclear. The many additional controls are also appreciated. As a suggestion, the authors may wish to revise their title to be more inclusive given the new data. However, I do see the newly added data seems more preliminary and this is understandable. Overall, I am quite pleased with the revision and would support publication. There remain a few small things to clarify, awkward phrasing, and organizational matters to address.

- I suppose due to various additions of new data, there were errors with referencing supplemental movies and figures as several of them were out of order. For example, supplemental movie 9 came early in the manuscript and supplemental figure 11 came before supplemental figures 9 and 10. Please go through and double check they are in order.
- Page 5, line 3: It was not immediately clear to me what expression lifespan meant in this context and it was later explained in the methods. It might be helpful to include a brief definition in the text.
- Page 7, line 4: should be 'Trace' instead of 'Tracing'.
- Page 8, line 11-13: not a great sentence, needs revision.
- I found the description in the section on 'Autonomously deforming liposomes by internally synthesized MinDE proteins' to be self-contradictory. The section has statement that says binding of MinD causes liposome to deform, and later states that it is actually the binding of MinE that contributes to liposome deformation. The authors can be more clear and precise on their description here.
- Page 16, line 15-16: this is not a great sentence since the purpose of the experiment is to visualize FtsZ and the use of 'mere' seems to trivialize the use of labeled protein that serves to precisely give the intended readout.
- Title of Figure 6 is not a true statement as I do not see how the experiments support the statement that cell-free expressed MinC enables coupling with FtsZ. This should be revised.
- Page 19, line 1-2: PURE system was shown in this study to co-express MinDE and FtsZ membrane-anchoring proteins, so 'might' is not accurate here.

Reviewer #3:

Remarks to the Author:

The authors have responded to the various comments by expanding the text to include more explanatory information and included new experiments.

I have one question about the new experiment in Fig. 6. How does this pattern compare to what is seen in Fig. 2B? Is the pattern of MinD/E/C affected by FtsA/Z? Is there a control where MinC is not added? I know that there is a control for the first part where there is no MinE (6a-just MinC and MinD). However, for part C what happens when MinC is omitted? Expect that it would look like 6A (left panel)?

I agree that Fig. 6 is novel since Schwille's group has tried this without FtsA (just ftsZ with a fused amphipathic helix (ref 30)). Some comparison of the results should be mentioned. One might expect the results presented here to be more representative of the native system (by using ftsA). In other words is FtsZ more responsive to the Min system?

Reviewer #4:

None

Second revision for:

Manuscript: "De novo synthesized Min proteins oscillate in liposomes and drive membrane deformation" by Godino et al.

Article reference: NCOMMS-19-14437A

Last correspondence: e-mail from 19 August, 2019

We thank Reviewers #1 and #3 for their enthusiastic evaluation of the revised manuscript and for their constructive comments that helped us further improve the manuscript. The Referee reports are in *blue text* and our point-by-point responses are in *black text*. Changes in the manuscript (main text only) are highlighted in *red text*.

We are pleased to provide you with a newly revised manuscript that addresses all remaining referees' questions.

Reviewers' comments:

Reviewer #1 (Remarks to the Author):

The revised manuscript has improved in several regards and the authors have taken care to address the main concerns that I raised previously. I was particularly impressed by the additional work that included FtsA and FtsZ in the present system, and I think this data adds significantly to the novelty of the science that was previously unclear. The many additional controls are also appreciated. As a suggestion, the authors may wish to revise their title to be more inclusive given the new data. However, I do see the newly added data seems more preliminary and this is understandable. Overall, I am quite pleased with the revision and would support publication. There remain a few small things to clarify, awkward phrasing, and organizational matters to address.

Reply: We greatly appreciate the positive feedback. The new experiments shown in Fig. 6d, e (as a request of Reviewer #3) and repeats of those shown in Fig. 6d, provide a solid dataset revealing, for the first time, the spatial regulation of FtsA-FtsZ patterns by MinDE(C). To capture these findings in the title, we propose “**De novo synthesized Min proteins drive oscillatory liposome deformation and regulate FtsA-FtsZ cytoskeletal patterns**” as the new title. Moreover, we modified the last sentence of the Introduction to be more inclusive of the latest results (page 4, starting on line 2).

- I suppose due to various additions of new data, there were errors with referencing supplemental movies and figures as several of them were out of order. For example, supplemental movie 9 came early in the manuscript and supplemental figure 11 came before supplemental figures 9 and 10. Please go through and double check they are in order.

Reply: We fixed the referencing.

- Page 5, line 3: It was not immediately clear to me what expression lifespan meant in this context and it was later explained in the methods. It might be helpful to include a brief definition in the text.

Reply: We added the definition: “**defined as the time points at which protein production stops**”.

- Page 7, line 4: should be ‘Trace’ instead of ‘Tracing’.

Reply: We changed it.

- Page 8, line 11-13: not a great sentence, needs revision.

Reply: We rephrased the sentence, now on page 9, lines 5-7.

- I found the description in the section on ‘Autonomously deforming liposomes by internally synthesized MinDE proteins’ to be self-contradictory. The section has statement that says

binding of MinD causes liposome to deform, and later states that it is actually the binding of MinE that contributes to liposome deformation. The authors can be more clear and precise on their description here.

Reply: We agree that the apposition of the two statements is confusing. We deleted the sentence that says that MinD binding to the bilayer increases the membrane surface area. On the one hand it cannot be the main factor; on the other, the reading is more fluid if the sentence is removed. In the last paragraph of this section (now on page 15, lines 6-15), we already discuss the possible roles of MinD and MinE binding to the membrane. We believe that the level of the discussion is sufficient, given the available data.

- Page 16, line 15-16: this is not a great sentence since the purpose of the experiment is to visualize FtsZ and the use of 'mere' seems to trivialize the use of labeled protein that serves to precisely give the intended readout.

Reply: We deleted 'mere'.

- Title of Figure 6 is not a true statement as I do not see how the experiments support the statement that cell-free expressed MinC enables coupling with FtsZ. This should be revised.

Reply: We used 'coupling' in the sense that it reorganizes FtsZ. We changed the title as: "Cell-free expressed MinDE(C) regulates FtsZ spatial organization".

- Page 19, line 1-2: PURE system was shown in this study to co-express MinDE and FtsZ membrane-anchoring proteins, so 'might' is not accurate here.

Reply: We have not expressed all proteins in one-pot reactions. MinC, MinDE and FtsA have been expressed separately and then mixed. Only MinD and MinE were co-expressed. Therefore, a future challenge consists to co-express all necessary proteins for FtsZ-ring positioning in liposomes. We substituted 'might' with 'should'.

Reviewer #3 (Remarks to the Author):

The authors have responded to the various comments by expanding the text to include more explanatory information and included new experiments.

I have one question about the new experiment in Fig. 6. How does this pattern compare to what is seen in Fig. 2B? Is the pattern of MinD/E/C affected by FtsA/Z?

Reply: There was no MinC in Fig. 2B. In the absence of FtsA/Z, traveling MinDE waves have a wavelength of $43 \pm 7 \mu\text{m}$ and a velocity of $0.49 \pm 0.06 \mu\text{m s}^{-1}$. We conducted new experiments that aimed at combining MinDE with FtsA/Z patterns (new Fig. 6d, e). Low-amplitude MinDE waves with wavelength $152 \pm 53 \mu\text{m}$ and velocity $2.8 \pm 0.9 \mu\text{m s}^{-1}$ were observed. Therefore, our data suggest that FtsA/Z cytoskeletal patterns might influence MinDE waves properties by increasing both the wavelength and velocity. The corresponding results and discussion points are reported on pages 17-18.

Is there a control where MinC is not added? I know that there is a control for the first part where there is no MinE (6a-just MinC and MinD). However, for part C what happens when MinC is omitted? Expect that it would look like 6A (left panel)?

Reply: We addressed this question in the newly revised manuscript (new Fig. 6d, e and new text on pages 17-18). Indeed, rings of FtsA/Z are observed just like in Fig. 6A (left) but superimposed traveling waves are also visible (see also new Supplementary Movie 11).

I agree that Fig. 6 is novel since Schille's group has tried this without FtsA (just ftsZ with a fused amphipathic helix (ref 30)). Some comparison of the results should be mentioned. One might expect the results presented here to be more representative of the native system (by using ftsA). In other words is FtsZ more responsive to the Min system?

Reply: We commented on this point on page 18, lines 1-4. To be more inclusive of our new results with FtsA/Z, we decided to modify the title, as suggested by Reviewer #1.

Reviewers' Comments:

Reviewer #4:

Remarks to the Author:

The authors resolved all my questions. I have no other comments.